# AdaReasoner: Dynamic Tool Orchestration for Iterative Visual Reasoning

**Mingyang Song**[*,1]**, Haoyu Sun**[*,2]**, Jiawei Gu**[*,3]**, Linjie Li**[*,4]**, Ranjay Krishna**[4,†]**, Yu Cheng**[5,†]

[1]Fudan University,  [2]Tongji University,  [3]National University of Singapore,
[4]University of Washington,  [5]The Chinese University of Hong Kong

🏠 Homepage: `https://adareasoner.github.io`
⬤ Code: `https://github.com/ssmisya/AdaReasoner`
🤗 Models and Data: `https://huggingface.co/AdaReasoner`

## Abstract

While augmenting Multimodal Large Language Models (MLLMs) with tools is a promising direction, current approaches face critical limitations. They often rely on single, atomic tools, failing to address the challenges of multi-turn planning, and they do not equip models with the ability to select effective tool combinations for complex tasks. To overcome these limitations, we introduce AdaReasoner, a framework that teaches models to perform dynamic tool orchestration for iterative visual reasoning. Our paradigm is designed to support a broad spectrum of tools, including computationally intensive, expert-model-based services. It features a comprehensive design that includes a new data curation methodology and a tailored Tool GRPO algorithm to optimize multi-turn tool-calling trajectories, which yields state-of-the-art models that achieve substantial gains over their baselines (+38.7% average on 7B) and reach near-perfect accuracy on challenging benchmarks like VSP (97.6%). This performance rivals or even surpasses leading proprietary models such as GPT-5 and Claude Sonnet 4, demonstrating that our approach can effectively overcome scale-based limitations by augmenting smaller models with powerful tool-use capabilities. Critically, we find that AdaReasoner develops emergent, self-adaptive behaviors: it learns to autonomously adopt beneficial tools, discard irrelevant ones, and modulate its usage frequency. This ability to curate its own optimal problem-solving strategies represents a significant step toward building more robust, scalable, and reliable reasoning agents.

## 1 Introduction

Multimodal LLMs have made steady progress on vision–language tasks, but a core challenge in multimodal reasoning remains. The problem lies in two areas: fine-grained perception and multi-step reasoning. On tasks such as visual spotting (Shu et al., 2025; Zhang et al., 2025a), models can often locate a relevant region but fail to capture the key details inside it. Without this evidence, their language skills become ungrounded and default to semantic priors, leading to "guided guessing": outputs that sound plausible but are brittle and detached from the image. The weakness is not in language generation itself, but in perception – the lack of iterative probing and refinement of visual understanding. Addressing this requires a shift from passive recognition toward structured reasoning and active manipulation of visual elements (Qi et al., 2024; Li et al., 2025a).

A promising direction for addressing this limitation is dynamic multimodal interaction (Lin et al., 2025), where the model iteratively refines visual states and reduces hallucinations. This aligns with the Extended Mind Theory (Clark & Chalmers, 1998), which views external tools as integral to cognition. For visual reasoning, tools should not be static add-ons, but active supports for manipulating and refining visual representations. Early SFT- and prompt-based methods (Ma et al., 2024; Hu et al., 2024) explored the use of multiple pre-defined tools, but typically relied on scripted invocation rather than active planning. More recent RL-based efforts, such as DeepEyes (Zheng et al.,

---

*Equal contribution.    †Equal Advisory Contribution

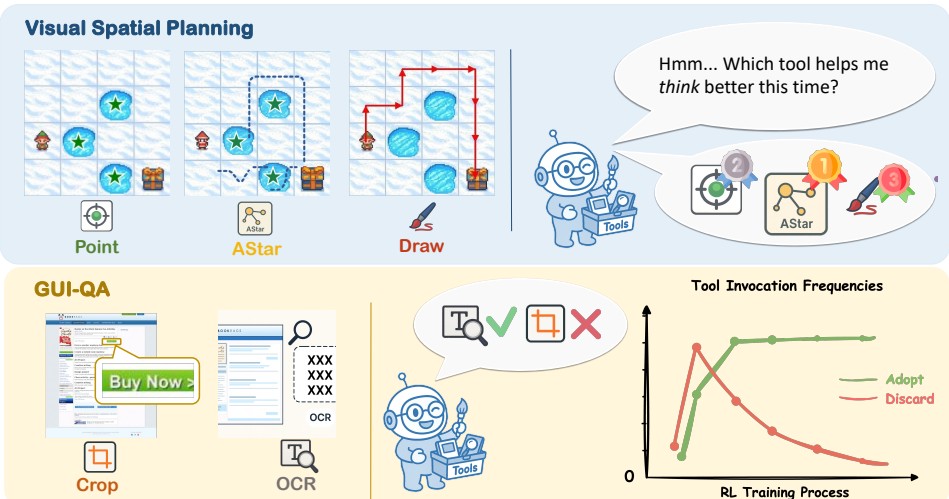

**Figure 1:** AdaReasoner adaptively selects the necessary tools to solve complex reasoning tasks. The model demonstrates the ability to acquire new tools, discard ineffective ones, and dynamically adjust the frequency of tool usage during both training and inference time within different tasks.

2025) and Pixel-Reasoner (Su et al., 2025b), enhanced perception through cropping-based search, yet restricted themselves to single-tool trajectories. Across both lines of work, what remains missing is the ability to plan, adaptively select, and coordinate tools—recognizing that deciding which tools to use, when to use them, and how to combine them is itself a critical form of multimodal reasoning.

We present **AdaReasoner**, a tool-aware reasoning agent that addresses the limitations of prior single-tool or scripted tool-use approaches. To bootstrap this learning, we introduce a new data curation pipeline that generates complex, multi-turn trajectories, explicitly modeling reflection and backtracking. This initial policy is then refined using our adaptive reinforcement learning paradigm, which is tailored to optimize these multi-turn, tool-planning strategies. We have also curated a high-quality dataset of multi-turn tool-use trajectories that incorporates a variety of sophisticated operations. AdaReasoner learns to adaptively plan and combine tools in multi-turn settings through cold-start and RL, following an iterative process of observing, manipulating, verifying, and reflecting. Our toolset supports both manipulation tools (e.g., DRAWLINE, INSERTIMAGE) and perception tools (e.g., POINT, OCR). It spans lightweight offline utilities as well as advanced model-based services. As illustrated in Figure 1, this design allows AdaReasoner to not only extract and check visual evidence but also actively transform it, yielding deeper multimodal reasoning.

Through adaptive tool interaction, AdaReasoner achieves substantial and stable gains across diverse benchmarks, with the 7B model improving by **+38.7%** on average and reaching near-perfect accuracy on tasks such as Visual Spatial Planning (**97.6%** vs. 31.6% baseline). It also surpasses proprietary systems, outperforming Claude Sonnet 4 on VSP (**97.6%** vs. 56.3%) and GPT-5 on Jigsaw (**96.6%** vs. 80.1%). Beyond accuracy, AdaReasoner demonstrates how tools shape reasoning: perception tools help models to see, manipulation tools help models to verify, and planning tools help models to calculate. Crucially, as shown in Figure 1, AdaReasoner exhibits *self-adaptive tool-use behaviors*. It learns to select effective tools, discard irrelevant ones, and regulate their use according to task demands and feedback, revealing strong flexibility and generalization. This addresses the long-standing question of which tools should be included and how models should learn to use them, suggesting that with proper training, MLLMs can autonomously curate tool-use strategies from a broad candidate set and extend their visual reasoning capacity in a goal-directed manner. In summary, our main contributions are as follows:

- We propose a comprehensive method for developing tool-augmented models, built upon three core innovations: a data curation method for multi-turn tool planning, an adaptive RL framework for multi-turn tool interaction, and a versatile tool suite supporting both lightweight tools and compute-heavy services.

- Based on our method, we introduce **AdaReasoner**, a new family of state-of-the-art models for complex tool planning, which develops emergent, self-adaptive behaviors, learning to autonomously **adopt** beneficial tools, **discard** irrelevant ones, and **modulate** its usage frequency.

- Our AdaReasoner achieves significant gains over their base counterparts and delivers performance that is competitive with, or superior to, leading proprietary models like GPT-5 and Claude Sonnet 4 on structured-reasoning tasks. This establishes that our methodology can elevate smaller, open-source models to the state-of-the-art.

## 2 RELATED WORK

### 2.1 REINFORCEMENT LEARNING FOR MULTIMODAL REASONING

The recent success of DeepSeek-R1 (Guo et al., 2025), which demonstrated that rule-based Group Relative Policy Optimization (GRPO) can effectively induce strong reasoning behaviors in LLMs, has spurred a wave of research aimed at replicating this paradigm in the multimodal domain. Several studies have successfully extended this approach, with Zhou et al. (2025) reproducing the emergent "aha" moment in MLLM reasoning, R1-OneVision (Yang et al., 2025a) introducing a cross-modal formalization pipeline, and works like Feng et al. (2025) and Li et al. (2025b) improving temporal reasoning in videos. A collection of other strong works have also leveraged R1-style methods to achieve impressive results in general MLLM reasoning (Huang et al., 2025; Shen et al., 2025; Lu et al., 2025). However, a key limitation of the R1-style, rule-based reward structure is that it primarily targets the reasoning process and does not directly improve the model's underlying perceptual abilities. Since accurate perception is the foundation for sound reasoning, error accumulation from faulty perception can still lead to hallucinations and degrade performance. AdaReasoner directly addresses this shortcoming. By leveraging the precise perceptual capabilities of external expert models and specialized tools, our framework ensures a high-fidelity understanding of the visual input, thereby improving the reliability of the entire reasoning pipeline.

### 2.2 TOOL-AUGMENTED MULTIMODAL REASONING

There is a growing interest in enhancing MLLMs with sophisticated tool-use capabilities. Early efforts focused on foundational aspects such as infrastructure and data. LLaVA-Plus (Liu et al., 2024a), for example, introduced a dedicated tool server to provide services for MLLMs. On the data front, CogCoM (Qi et al., 2024) identified six key manipulation strategies and trained models on synthetic Chain-of-Manipulation (CoM) data, while TACO (Liu et al., 2024b) contributed a large-scale dataset of reasoning traces derived from 15 visual tools. Subsequent research has explored different paradigms for tool interaction. One prominent line of work enhances visual reasoning by training models to generate code (Zhang et al., 2025b; Zhao et al.). While powerful, these code-based environments are ill-suited for integrating computationally intensive capabilities, such as invoking large expert models. Another line of research leverages simpler, atomic visual tools like zoom-in functions to augment model perception (Wang et al., 2025; Zheng et al., 2025; Su et al., 2025a; Zhu et al., 2025b; Su et al., 2025c). However, these approaches typically focus on single-step actions and have not explored the more complex challenges of multi-turn planning or dynamic tool composition. Our work, AdaReasoner, is designed to bridge these gaps, providing a framework that enables models to perform multi-turn planning and reasoning while adaptively selecting from a diverse suite of tools.

## 3 METHOD

### 3.1 PRELIMINARY

**Problem Formulation** As shown in Figure 2, we formalize tool-augmented multimodal reasoning as a sequential decision-making process. An MLLM represented as a policy $\pi_\theta$ parameterized by weights $\theta$, is tasked with solving a problem by generating a reasoning trajectory $\tau$. The policy is equipped with access to a predefined set of visual tools $T = \{t_1, \ldots, t_n\}$.

A trajectory $\tau$ is a sequence of state-action-observation tuples that represent the model's step-by-step reasoning process:

$$\tau = \{(s_0, a_0, o_0), (s_1, a_1, o_1), \ldots, (s_T, a_T, o_T)\} \tag{1}$$

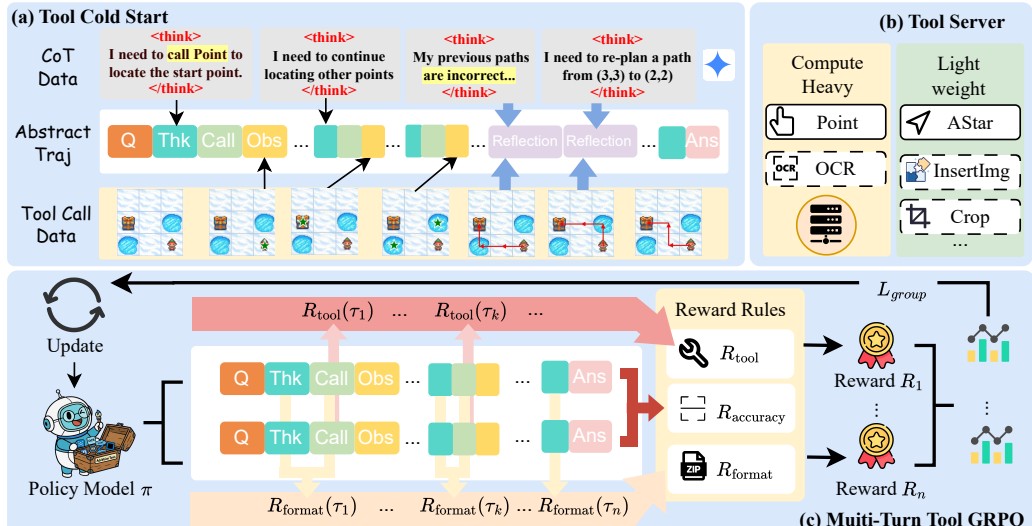

**Figure 2:** An overview of our AdaReasoner framework. The process consists of two stages: (a) a Cold Start phase, where the trajectory is specially designed for multi-turn reasoning, and (c) a Tool GRPO phase, where the policy is refined via reinforcement learning guided by our adaptive, multi-turn reward. The central Tool Server (b) manages a diverse suite of both lightweight and compute-heavy tools, enabling all interactions throughout the pipeline.

Here, $s_t$ denotes the problem state, $a_t \in \mathcal{T}$ is a tool-calling action encapsulated by special tokens, and $o_t$ is the resulting observation from the tool's execution. Each action $a_t$ induces a transition from state $s_t$ to $s_{t+1}$ based on the new information in $o_t$:

$$s_0 \xrightarrow{a_0} s_1 \xrightarrow{a_1} s_2 \xrightarrow{a_2} \dots \xrightarrow{a_T} s_{T+1} \tag{2}$$

**Visual Tools** Our AdaReasoner framework is built upon a diverse and powerful suite of visual tools, which it executes and integrates directly into the reasoning process. This toolset is intentionally designed to cover three core reasoning functions: **perception** (e.g., POINT, OCR), **manipulation** (e.g., DRAWLINE, INSERTIMAGE), and **calculation** (e.g., ASTAR). Furthermore, this suite seamlessly integrates both lightweight, offline tools for immediate execution and computationally intensive, expert-model-based online tools. These foundational capabilities are summarized in Table 1, with detailed specifications for each tool provided in Appendix B.1.

## 3.2 HIGH-QUALITY TRAJECTORY DATA CURATION

As illustrated in Figure 2a, our data curation follows a unified, three-stage process designed to generate high-fidelity, human-like reasoning trajectories.

**Abstract Trajectory Design** First, for each task, we manually design an abstract, optimal problem-solving blueprint. For example, the **VSP** trajectory follows a perception-planning-verification logic, **Jigsaw** mimics an iterative trial-and-error process, and **GUIQA** involves a focus-then-extract strategy. However, to ensure the model develops true robustness beyond simply following these "perfect" paths, we deliberately incorporate two critical types of complex scenarios:

- **Reflection and Backtracking:** We include trajectories designed to encourage a process of trial and verification. These feature explicit self-correction steps where the model must reflect on a sub-optimal outcome and backtrack, teaching it to actively validate its own hypotheses and learn from intermediate failures.

- **Explicit Tool Failure:** To prevent over-reliance on external tools, we introduce cases where tools fail or return erroneous results. In these scenarios, after recognizing that a tool is not providing a useful output, the trajectory prompts the model to fall back on its own intrinsic capabilities to generate a "best-effort" answer, ensuring it develops a resilient, dual-strategy approach.

**Table 1:** Visual tools integrated within AdaReasoner. We illustrate their arguments, outputs, and core functions description. More detailed descriptions of our tools are presented in Appendix B.1.

| Tool | Description | Arguments | Tool Output |
|---|---|---|---|
| POINT | Point to a target object | Image + Description | Point coordinates |
| DRAW2DPATH | Draw a path using directional commands | Image + Start + Directions | Image with a line |
| ASTAR* | Use A* to find the shortest obstacle-free path | Start + Goal + Obstacle | Shortest path |
| DETECTBLACKAREA | Detect pure black areas in an image | Image | Bounding boxes of black areas |
| INSERTIMAGE | Insert image into base at bounding box position | Image + Coordinates + Insert | Combined image |
| OCR | Extracts and localizes text from the image | Image | Text with their bounding box |
| CROP | Crop a region and augment it | Image + Coordinates | Cropped Image |

**Tool Calling Supplements**  Subsequently, we ground these abstract blueprints by programmatically executing the tool calls to populate them with concrete, real-world inputs and outputs.

**CoT Data Generation**  Finally, we leverage a powerful LLM to generate the corresponding Chain-of-Thought (CoT) reasoning that connects each step. This process yields a final dataset of rich, tool-augmented trajectories that teach the model not just *what* tools to call, but *why* and *how* to reason between them. Details for our trajectory data curation can be found in Appendix B.2.

### 3.3 MULTI-TURN TOOL GRPO

To train our model for complex multi-turn tool-planning scenarios, we extend the GRPO framework to effectively handle multi-turn tool-calling reasoning trajectories. Concretely, we use **Multi-turn Reward Accumulation** and **Adaptive Tool Reward** to ensure the efficacy of the RL procedure.

**Multi-turn Reward Accumulation** Our total reward, $R_{\text{total}}$ is formulated as $R_{\text{total}} = R_{\text{format}} \cdot (\lambda_{\text{tool}} \cdot R_{\text{tool}} + \lambda_{\text{acc}} \cdot R_{\text{acc}})$, with each component adapted for multi-turn trajectories $\tau = \{\tau_0, \ldots, \tau_T\}$.

- **Format Reward** $R_{\text{format}} = \prod_{i=1}^{n} R_{format}(\tau_i)$ Correct formatting is mandatory at every step. Therefore, the overall format reward for a trajectory is set to 1 if and only if every individual step within it is correctly formatted. A single format error at any turn results in $R_{\text{format}} = 0$, nullifying the entire reward for the trajectory. This enforces strict adherence to the reasoning structure.

- **Tool Reward**  The overall tool reward is the average of the fine-grained scores from all tool-calling turns (from $\tau_0$ to $\tau_{T-1}$). It is calculated as $R_{\text{tool}} = \frac{1}{T} \sum_{t=0}^{T-1} R_{\text{tool}}(\tau_t)$. Each individual tool call, $R_{\text{tool}}(\tau_t)$, is evaluated using a hierarchical score of 0-4 based on four criteria (Structure, Name, Parameter Name, and Parameter Content).

- **Accuracy Reward** This reward is granted only based on the final turn, $\tau_T$. If the final answer is correct, $R_{\text{acc}} = 1$; otherwise, it is 0. $\lambda_{\text{acc}}$ and $\lambda_{\text{tool}}$ are hyperparameters, the effects of which are analyzed in detail in Section 5.

**Adaptive Reward for Encouraging Tool Use.** To guide the model to use tools as a reliable aid when uncertain, we introduce an adaptive reward mechanism with an asymmetric incentive structure, where the reward calculation is contingent on the final answer's correctness. Correct trajectories automatically receive the maximum possible reward (8 points), irrespective of whether tools were used, thereby rewarding efficient solutions (including forgoing tools when unnecessary). Conversely, for incorrect trajectories, the reward is calculated component-wise. This creates a powerful safety net that trajectories with proper tool use can still earn partial credit (up to 4 points), while those that forgo tools and guess incorrectly are heavily penalized with zero reward. This design teaches the model that while direct answers are optimal when confident, a structured, tool-assisted process is the superior strategy when facing uncertainty. (See Appendix B.4 for details).

## 4 EXPERIMENTS

### 4.1 EXPERIMENT SETTINGS

**Models**  Our core experiments are conducted on the Qwen2.5-VL-3B-Instruct and Qwen2.5-VL-7B-Instruct models (Bai et al., 2025). These models are selected as our primary testbeds due to their strong open-source performance in visual understanding, allowing us to effectively demonstrate the impact and scalability of our proposed methods across different model sizes.

**Table 2:** Our main results on VSPO, VSP, Jigsaw, BLINK-J, GUIChat, and WebMMU benchmarks. TC, TG means Tool Cold Start and Tool GRPO, respectively. The best performance is highlighted in **bold**, while the second-best performance is indicated with an underline.

| Model | VSPO | | | VSP | | | Jigsaw | BLINK-J | GUIChat | WebMMU | | | |
|---|---|---|---|---|---|---|---|---|---|---|---|---|---|
| | Nav | Verify | Overall | Nav | Verify | Overall | | | | Avg. | Act. | Compre. | Rea. |
| Qwen 2.5 VL 32B | 7.56 | 53.12 | 28.56 | 24.33 | 45.40 | 33.91 | 59.50 | 64.67 | 85.21 | 71.27 | 85.98 | 68.65 | 61.82 |
| Qwen 2.5 VL 72B | 17.22 | 52.34 | 33.41 | 28.00 | 52.40 | 39.09 | 70.10 | 71.33 | 88.01 | **77.10** | **91.06** | 74.59 | 68.14 |
| InternVL3 78B | 7.22 | 52.60 | 28.14 | 21.67 | 51.20 | 35.09 | 52.80 | 60.00 | 79.83 | 62.47 | 71.34 | 73.27 | 51.25 |
| GPT 5 | 26.89 | 42.86 | 34.25 | 48.17 | 64.60 | 55.64 | 80.10 | 73.33 | 71.41 | 62.13 | 80.49 | 68.65 | 45.96 |
| Gemini 2.5 flash | 15.44 | 68.96 | 40.12 | 34.50 | 76.40 | 53.55 | 67.20 | 65.33 | 83.05 | 69.31 | 66.26 | 73.93 | **69.46** |
| Claude 4 sonnet | 37.56 | 67.92 | 51.56 | 48.17 | 66.00 | 56.27 | 58.60 | 65.33 | **93.14** | 71.61 | 83.54 | **77.23** | 60.50 |
| Qwen2.5 VL 3B | 5.67 | 50.91 | 26.53 | 7.50 | 49.80 | 26.73 | 39.80 | 48.67 | 45.11 | 45.39 | 55.89 | 51.82 | 34.95 |
| + Direct SFT | 27.42 | 49.66 | 38.15 | 34.50 | 44.00 | 38.82 | 42.60 | 53.33 | 55.51 | 46.54 | 61.38 | 54.46 | 32.31 |
| + Direct GRPO | 2.78 | 50.00 | 24.55 | 18.33 | 50.00 | 32.73 | 42.70 | 52.67 | 52.49 | 48.44 | 56.30 | 51.49 | 41.41 |
| + Our TC | 14.67 | 84.81 | 47.01 | 23.33 | 84.40 | 51.09 | 66.00 | 70.00 | 45.32 | 35.03 | 44.72 | 42.24 | 24.82 |
| + Our TG | 11.22 | 50.00 | 29.10 | 22.67 | 50.00 | 35.09 | 43.00 | 47.33 | 89.60 | 58.88 | 72.15 | 62.05 | 47.87 |
| + Our TC + TG | 73.00 | 98.44 | 84.73 | 92.17 | 97.80 | 94.73 | 94.80 | 88.67 | 85.45 | 63.48 | 81.71 | 57.43 | 53.01 |
| Δ | +67.33 | +47.53 | +58.20 | +84.67 | +48.00 | +68.00 | +55.00 | +40.00 | +40.34 | +18.09 | +25.82 | +5.61 | +18.06 |
| Qwen2.5 VL 7B | 9.84 | 50.85 | 29.62 | 14.17 | 52.60 | 31.64 | 45.70 | 52.67 | 59.46 | 62.67 | 77.03 | 69.64 | 49.19 |
| + Direct SFT | 33.68 | 51.30 | 42.18 | 42.67 | 51.40 | 46.64 | 86.40 | 88.00 | 62.68 | 55.62 | 65.65 | 63.70 | 44.79 |
| + Direct GRPO | 10.33 | 49.48 | 28.38 | 12.50 | 51.40 | 30.18 | 64.90 | 80.00 | 67.67 | 70.19 | 83.54 | 69.31 | 60.94 |
| + Our TC | 31.58 | 94.01 | 61.69 | 41.00 | 93.60 | 64.91 | 84.20 | 83.33 | 61.85 | 51.63 | 64.63 | 54.13 | 41.12 |
| + Our TG | 65.89 | 52.47 | 59.70 | 88.17 | 55.20 | 73.18 | 72.30 | 80.67 | 92.52 | 72.97 | 88.62 | 66.34 | 64.61 |
| + Our TC + TG | **73.44** | **98.70** | **85.09** | **96.33** | **99.20** | **97.64** | **96.60** | **96.00** | 88.57 | 68.16 | 82.32 | 67.33 | 58.30 |
| Δ | +63.60 | +47.85 | +55.47 | +82.17 | +46.60 | +66.00 | +50.90 | +43.33 | +29.11 | +5.49 | +5.29 | -2.31 | +9.11 |

**Baselines** We benchmark our approach against a comprehensive set of baselines. (1) SOTA Proprietary Models: GPT-5-20250807 (OpenAI, 2025), Claude-sonnet-4-20250514 (Anthropic, 2025), and Gemini-2.5-flash (Comanici et al., 2025) (2) Competitive Open-Source MLLMs: Qwen-2.5-VL-32/72B-Instruct (Bai et al., 2025) and InternVL-3-78B (Zhu et al., 2025a). (3) Direct SFT: We take base models supervisedly finetuned on the training set of each task as a strong baseline (Yang et al., 2025b). (4) Direct GRPO: Following prior work (Zhou et al., 2025), we apply rule-based GRPO to the base models to enhance their reasoning ability, serving as another strong baseline.

**Tasks** Our approach is evaluated across three diverse tasks designed to probe distinct facets of multimodal reasoning: **Visual Spatial Planning**, for multi-step planning and perception, evaluated on our custom out-of-distribution benchmark (VSPO) and the standard VSP benchmark (Wu et al., 2024). **Jigsaw**, for visual compositionality, evaluated on our Jigsaw-COCO dataset and the Jigsaw subset from BLINK (Fu et al., 2024) and **GUIQA**, for fine-grained GUI understanding, evaluated on GUIChat (Chen et al., 2024) and WebQA from the WebMMU benchmark (Awal et al., 2025). Detailed settings and implementation details for all tasks are provided in Appendix C.1.

## 4.2 BRIDGING PERCEPTION AND REASONING VIA VISUAL TOOLS

**AdaReasoner could bring stable improvements** As shown in Table 2, our AdaReasoner framework consistently and dramatically improves the performance of base models, demonstrating an average gain of **+38.66%** on the 7B model. This tool-augmented approach transforms tasks like VSP from a low baseline (∼31.64%) to near-perfect execution (**97.64%**). This performance significantly surpasses traditional optimization methods such as task-specific SFT (**46.64%**) and Direct GRPO (**30.18%**). Furthermore, AdaReasoner enables our 7B model to achieve state-of-the-art results that are competitive with, and in structured-reasoning domains, superior to the best proprietary models. For instance, on VSP and Jigsaw, our model outperforms Claude Sonnet 4 (**97.64% vs. 56.27%**) and GPT-5 (**96.60% vs. 80.10%**) respectively. This confirms that AdaReasoner is a highly effective strategy for unlocking advanced reasoning capabilities in open-source models.

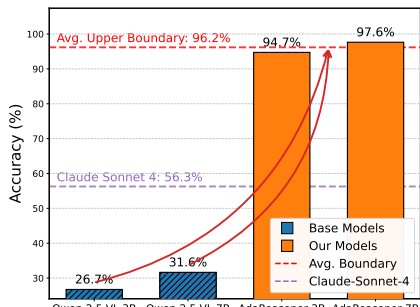

**Figure 3:** Overcoming scale-based limitations with tool augmentation. On the VSP task, our tools boost the performance of both 3B and 7B models, elevating them from disparate baselines to a near-uniform high performance.

**AdaReasoner could help overcome scale-based limitations**    Furthermore, our results reveal that tool augmentation can redefine the performance ceiling of MLLMs by overcoming scale-based limitations. As illustrated in Figure 3, while the baseline performance of 3B and 7B models is disparate and low, our tool-augmented versions both achieve near-perfect accuracy (94.7% and 97.6%). This strongly indicates that the primary performance bottleneck has shifted from the model's intrinsic scale to the extrinsic quality of the tools it wields. Consequently, this establishes a powerful paradigm where even smaller, more efficient models can achieve state-of-the-art results, contingent not on their size, but on the instruments they are equipped with.

## 4.3   TOOLS HELP MLLMs TO SEE, VERIFY AND PLAN

Our framework decomposes complex reasoning tasks into manageable steps, each resolved either by the model itself or by a high-precision external tool. This design fundamentally shifts the problem-solving burden: instead of requiring flawless internal reasoning, the model's primary task becomes effective tool planning. By delegating precise sub-tasks to reliable tools, the model is freed to focus on its core competencies of judgment, synthesis, and integrating the resulting outputs.

**Perception Tools Help MLLMs to See**   Our framework leverages expert perception tools to overcome the intrinsic perceptual limitations of MLLMs. As shown in Table 3 and 4, in VSP-verification, our expert POINT tool achieves perfect localization accuracy (100.0% vs. ~50.0% for baselines), and providing its coordinate output as context boosts the downstream zero-shot reasoning performance by an average of **+18.79** points. This principle holds even with imperfect tools: for the Jigsaw task,

**Table 3:** Comparison of start-point localization accuracy between Molmo-7B-D (Deitke et al., 2024) and the Qwen-VL series base models.

| Model | Accuracy |
|---|---|
| Qwen 2.5 VL 3B Instruct | 2.47 |
| Qwen 2.5 VL 7B Instruct | 47.01 |
| Qwen 2.5 VL 32B Instruct | 6.54 |
| Qwen 2.5 VL 72B Instruct | 50.0 |
| **Our POINT Tool** (Deitke et al., 2024) | **100.0** |

our DETECTBLACKAREA tool provides a robust **72.6%** accuracy, offering a significant perceptual advantage that underscores the value of delegating these challenges to specialized tools.

**Manipulating Tools help MLLMs to verify**
Our manipulating tools empower the model to formulate and subsequently verify its own hypotheses. For example, in the VSP-Verify task, we teach the model to call DRAW2DPATH to explicitly draw a red line on the frozen lake question picture. The problem is thus converted to verifying whether the red line crosses the blue ice holes. As shown in Table 4, even under a zero-shot context-appending setting, the

**Table 4:** Impact of tool-augmented context on zero-shot reasoning accuracy for VSP-Verify task.

| Model | VSP-Verify | | |
|---|---|---|---|
| | Base | /w Line | /w Point |
| Qwen 2.5 VL 3B | 50.91 | 57.92 (+7.01) | 49.09 (-1.82) |
| Qwen 2.5 VL 7B | 50.85 | 57.68 (+6.83) | 57.87 (+7.02) |
| Qwen 2.5 VL 32B | 53.12 | 61.31 (+8.19) | 87.87 (+34.75) |
| Qwen 2.5 VL 72B | 52.34 | 61.57 (+9.23) | 87.53 (+35.19) |

DRAWLINE does help improve the judge accuracy of the model, yielding an average performance improvement of **+7.82** points. Similarly, in the Jigsaw task, the model can actively invoke the INSERTIMAGE tool to compose and evaluate different candidate pieces by inserting them into the original image, enabling more informed decision-making.

**High-Quality Trajectory data help MLLMs to plan**   While augmenting context with tool outputs is effective for zero-shot reasoning, this strategy alone is insufficient for achieving optimal performance. Tool-Cold-Start addresses this gap by explicitly teaching the model two foundational capabilities: how to use tools correctly and how to recognize the patterns where they should be applied. As shown in Table 2, for the 7B models, adding the Tool-Cold-Start phase before Tool-GRPO yields a massive performance improvement of **+24.93** points on VSP and **+19.82** points on Jigsaw compared to using Tool-GRPO alone. Besides this, the inclusion of reflection data during the Cold-Start phase provides further benefits to the model's reasoning. As shown in Table 5, when A* search is disabled, training with reflection data yields a substantial improvement over the no-reflection checkpoints (91.36 vs. 67.27).

**Table 5:** Adaptability study on the VSP and VSPO tasks. Stage compares our full Tool Cold Start (TC) + Tool GRPO (TG) pipeline against TC alone. Reflection indicates training with (✓) or without (✗) reflection data. A* specifies tool availability: during Reinforcement Learning (RL), at Inference (Inf), or unavailable (-). A* Statistics report calls per sample and success rate.

| Stage | Reflection | A* | VSP | | | VSPO | | | A* Statistics | |
|---|---|---|---|---|---|---|---|---|---|---|
| | | | Nav | Verify | Overall | Nav | Verify | Overall | CPS | Succ Rate |
| TC + TG | ✗ | RL | **96.33** | 99.20 | **97.64** | 73.44 | 98.70 | **85.09** | 0.56 | **100.00** |
| TC + TG | ✓ | - | 84.33 | **99.80** | 91.36 | 63.89 | **99.61** | 80.36 | 0.00 | 0.00 |
| TC + TG | ✓ | Inf | 55.17 | 84.60 | 68.55 | 57.22 | **99.61** | 76.77 | 0.68 | 16.89 |
| TC + TG | ✗ | Inf | 62.33 | 80.00 | 70.36 | 43.78 | 88.70 | 64.49 | 0.52 | 94.53 |
| TC Only | ✗ | - | 41.00 | 93.60 | 64.91 | 31.58 | 94.01 | 61.69 | 0.00 | 0.00 |
| TC + TG | ✗ | - | 44.83 | 94.20 | 67.27 | 27.67 | 94.81 | 58.62 | 0.00 | 0.00 |
| TC Only | ✗ | Inf | 46.00 | 79.40 | 61.18 | 32.11 | 81.43 | 54.85 | **0.49** | 85.16 |

### 4.4 MLLMs Can Learn Adaptive Tool-Using

To investigate whether MLLMs can effectively learn to select tools and adaptively regulate their usage frequency, we carried out a systematic study to build an adaptive tool planning model, which is the main characteristic of our AdaReasoner.

**MLLMs Can Use New Tools during Inference Time** During inference, the model demonstrates a remarkable ability to generalize, dynamically adapting its tool-use strategy to solve novel problems. To probe this capability, we investigated whether the model could leverage a powerful new tool, ASTAR, that was intentionally withheld during the Cold Start phase. As shown in Table 5, when the A* tool is introduced solely at inference time (Inf), it provides a significant performance boost to the relevant task. For our standard CS+GRPO model (without reflection), this elevates the VSP navigation score from 44.83 (without A*) to 62.33. The A* Statistics corroborate this adaptive behavior, showing a high invocation success rate of 94.53%, which indicates the model is not just guessing but is correctly learning the tool's syntax and purpose in a zero-shot setting. However, this adaptability also reveals a critical challenge. The presence of the new A* tool, which is irrelevant for the verification task, acts as a distractor and degrades performance. For the same model, the Verify score drops from 94.20 to 80.00 when the tool is made available. Notably, models trained with Reflection data appear to develop a more rigid policy, as they fail to effectively incorporate the new tool, leading to a significant performance decrease in navigation.

**Learning to Adopt and Master Beneficial Tools through RL** To increase the stability of the adaptive tool calling, we utilize RL for teaching the model to not just use a new tool, but to master its application context. We start from the same SFT checkpoint that has never been exposed to the A* tool. We then introduce A* as an available option during our tool GRPO procedure. The results, shown in Figure 4 and Table 5, are compelling. The key findings are as follows.

- **Learning to Adopt Beneficial Tools** As illustrated in Figure 4a, for the Path Navigation task (warm-colored curves), the model's invocation frequency for ASTAR progressively increases, stabilizing at a high rate of over 1.0 call per sample. This upward trajectory indicates that the model, guided by task-completion rewards, correctly identifies ASTAR as a highly beneficial tool for pathfinding and actively incorporates it into its problem-solving strategy. As a result, this mastery translates to a dramatic performance increase, with the VSP navigation score soaring to 96.33, which achieves the best performance.

- **Learning to Discard Irrelevant Tools** Second, and just as critically, the model learns to discard the tool when it is irrelevant. Figure 4a (cool-colored curves) shows the inverse trend for the Verification task. The model initially explores using the A* tool but, receiving no reward for doing so, gradually learns to suppress its usage, with the invocation frequency decaying towards zero. This adaptive pruning prevents the negative interference observed in the zero-shot inference scenario, allowing the Verification performance to remain at a near-perfect 99.20. Similar phenomena can be found in the GUIQA task as well, where the crop tool is discarded by the model under pure Tool-GRPO settings. This demonstrates that the model learns to suppress the usage of irrelevant tools, avoiding the potential interference or inefficiency they might introduce.

- **Learning to Modulate Tool-Use Frequency** Beyond the binary choice of adopting or discarding a tool, the model exhibits a more nuanced capability: dynamically modulating the invocation

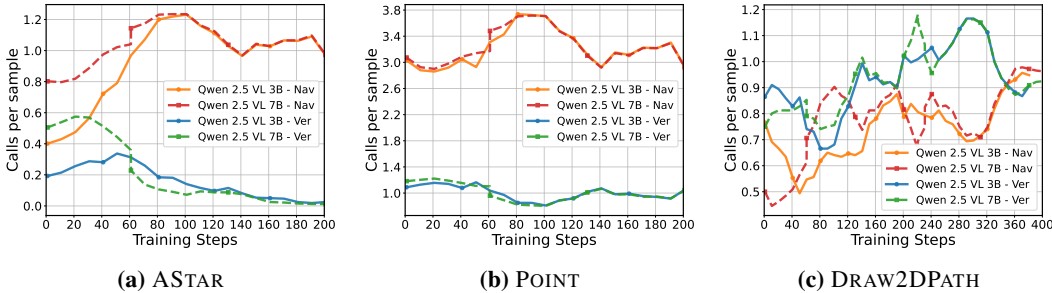

**(a)** ASTAR        **(b)** POINT        **(c)** DRAW2DPATH

**Figure 4:** Evolution of tool invocation frequencies for ASTAR, POINT, and DRAW2DPATH during reinforcement learning. The model is optimized on VSP Verification (cool-colored curves) and VSP Navigation (warm-colored curves) tasks.

frequency of continuously useful tools to find an optimal balance. This is evident in the usage patterns of **Point** and DRAW2DPATH (Figures 4b and 4c). For instance, in Figure 4b, the model learns that the Point tool is significantly more critical for navigation, maintaining a high and stable call frequency ($\sim 3.2$ calls/sample), while keeping its usage minimal for verification ($\sim 1.0$ call/sample). Similarly, for DRAW2DPATH, the model converges towards a moderate and stable invocation frequency for both tasks after an initial period of exploration. This behavior suggests that the tool-aware GRPO procedure enables the model to self-correct and fine-tune its tool-use strategy, converging on task-specific, efficient invocation patterns.

## 5 ABLATION STUDY

We systematically adjust $\lambda_{\text{tool}}$ and $\lambda_{\text{acc}}$ to evaluate their influence on learning dynamics and final performance. Specifically, we train the model on the same VSP task data for 100 RL steps under different reward-weight settings, monitor the training curves to ensure convergence, and then evaluate each checkpoint's performance. The results are summarized in Table 6.

As shown in Table 6, the model's performance consistently improves as the ratio $\lambda_{\text{tool}} : \lambda_{\text{acc}}$ increases. This indicates that larger tool rewards not only accelerate convergence during RL training but also lead to significantly better final performance. These results validate that our tool-reward design is effective and plays a crucial role in helping the model learn tool calling more efficiently and robustly.

**Table 6:** Ablation on reward-weight configurations for VSP and VSPO.

| $\lambda_{\text{tool}} : \lambda_{\text{acc}}$ | VSP (%) | | | VSPO (%) | | |
|---|---|---|---|---|---|---|
| | Nav | Verify | Overall | Nav | Verify | Overall |
| **0:1** | 51.83 | 95.00 | 71.45 | 41.78 | 75.58 | 57.37 |
| **1:2** | 49.50 | 95.80 | 70.55 | 36.44 | 94.29 | 63.11 |
| **1:1** | 64.00 | 96.40 | 78.73 | 48.56 | 96.23 | 70.54 |
| **2:1** | **90.33** | **96.80** | **93.27** | **70.33** | **96.36** | **82.34** |

## 6 QUALITATIVE RESULTS

Our qualitative analysis is shown in Figure 5, which reveals that AdaReasoner-7B's superior performance stems from its robust, process-oriented methodology, in contrast to the brittle, monolithic approach of baselines like GPT-5. This methodology manifests in three key capabilities. First, it performs multi-turn tool planning and reasoning, breaking down complex tasks like VSP-Navigation into a logical sequence of perception, planning, and verification. Second, it exhibits reflection, as seen in the Jigsaw task, where it actively evaluates imperfect tool outputs ("neither insertion perfectly resolves...") and adjusts its strategy from simple trial-and-error to a more nuanced comparative judgment. Finally, it demonstrates synergistic tool use; in GUIQA, it strategically combines Crop and OCR, using the former to create an ideal, unambiguous input for the latter, enabling focused and accurate information extraction.

## 7 CONCLUSION

We introduced AdaReasoner, a training framework combining Tool-SFT and RL that teaches models not just to use tools, but to compose them in a dynamic, task-aware manner. Our experiments show

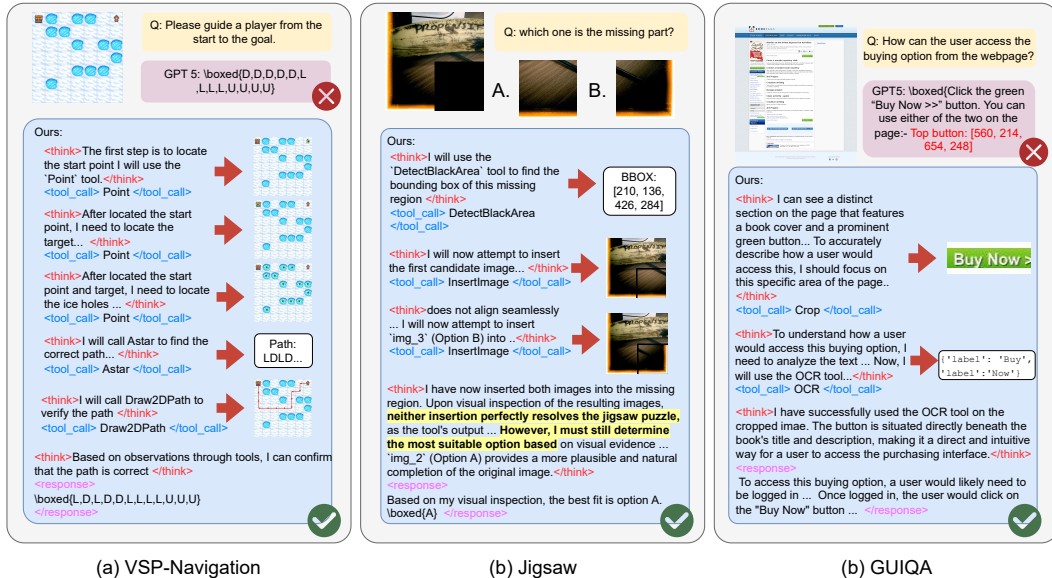

(a) VSP-Navigation        (b) Jigsaw        (b) GUIQA

**Figure 5:** Our AdaReasoner-7B demonstrates advanced capabilities for multi-turn, tool-assisted reasoning and reflection, enabling it to achieve performance that is on par with, or even superior to, state-of-the-art closed-source models.

that this approach leads to state-of-the-art performance and, more importantly, endows models with adaptive capabilities: they learn to adopt, discard, and modulate tool use as needed. Our central finding is that by effectively leveraging tools, the primary barrier to performance shifts from the model's inherent scale to the tool's external accuracy. This paradigm enables smaller models to achieve performance previously attainable only by larger models.

## 8    ETHICS STATEMENT

This research adheres to the ICLR Code of Ethics. Our work aims to enhance the reasoning capabilities of MLLMs through tool use, and we are committed to scientific transparency by detailing our methodology and planning to release our training and evaluation framework to encourage reproducibility. We acknowledge the potential for dual-use, as more capable agents could be misused. The outputs of AdaReasoner may reflect biases from its base models and the tools it uses. All data used is either procedurally generated or from public benchmarks, with no private user data collected. We believe the benefits of enabling smaller models to perform complex reasoning and advancing the understanding of agentic AI are significant, and we encourage continued research into the safety and alignment of such systems.

## 9    REPRODUCIBILITY STATEMENT

Detailed descriptions of our methodology, covering both the Tool Cold Start and Tool GRPO stages, are provided in Appendix B. In addition, comprehensive experimental settings, including all hyperparameters, are documented in Section 4.1 and further elaborated in Appendix C.3. To ensure full reproducibility and to allow others to build upon our work, we will publicly release our complete source code—covering the entire framework for data curation, training, and evaluation—along with all custom-generated data upon publication.

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

# A   USE OF LARGE LANGUAGE MODELS

In adherence to the ICLR policy on the use of Large Language Models (LLMs), we explicitly state that LLMs were not used to generate any of the core ideas, methodologies, or experimental results presented in this paper. The conceptualization of the AdaReasoner framework, the design of the training pipeline, and the analysis of the results were conducted entirely by the human authors. We did, however, utilize LLM-based tools (such as Google's Gemini) for tasks related to improving the manuscript's language, clarity, and grammatical correctness, akin to using a spelling or grammar checker. All final text was reviewed and edited by the authors to ensure it accurately reflects our original contributions and findings.

# B   METHOD DETAILS

## B.1   BASIC SETTINGS

We first formalize multimodal reasoning with tools as an agentic planning process, enabling a systematic description of how models decompose and solve complex tasks.

**Problem Formulation**   We denote an MLLM with tool-using capability as a policy $\pi_\theta$, parameterized by model weights $\theta$. At initialization, $\pi_\theta$ is equipped with access to a pool of tools $\mathcal{T} = t_1, t_2, \cdots, t_n$, where $n$ denotes the number of available tools. Given a task description $g$ and the original multimodal input $x = \text{text}, \text{image}$, the system begins from an initial state $s_0$. Building on this setup, the planning framework is formalized through three essential components. **State** $s_t$ represents the current problem status: the initial state $s_0$ corresponds to the original input, while intermediate states capture textual reasoning steps conditioned on accumulated observations, until a special token triggers an action. **Action** $a_t$ denotes a one-step tool invocation, delimited by the special symbols `<tool_call>` and `</tool_call>`, which executes a selected tool. **Observation** $o_t$ is the execution result returned by the invoked tool and is incorporated into the subsequent state.

A typical tool-integrated reasoning trajectory $\tau$ involves multiple tool invocations over several reasoning steps, which can be represented as a sequence of rounds:

$$\tau = \{\tau_0, \tau_1, \tau_2, \ldots, \tau_T\}$$

where each round is defined as $\tau_i = \{s_i, a_i, o_i\}$, and the sequence proceeds as follows:

$$s_0 \xrightarrow{a_0} s_1 \xrightarrow{a_1} s_2 \xrightarrow{a_2} \ldots \xrightarrow{a_T} s_{T+1}$$

To enable the model to autonomously generate reasoning traces and tool calls, we utilize a system prompt as shown in C.2 during rollout. The tool list placeholder denotes the tool set $\mathcal{T}$, which contains all tools available for invocation.

**Tool Definition and Usage**   This section provides a detailed description of the visual tools integrated within our AdaReasoner framework. For each tool, we outline its core functionality, input arguments, output format, and its specific role in addressing the challenges of our evaluation tasks.

- **POINT**
  - **Functionality:** A perceptual tool designed for precise object localization. Given an image and a natural language description of a target (e.g., "the start point", "red cars"), it returns a list of pixel coordinates (x, y) of the objects' center.
  - **Role in VSP:** This tool is fundamental for grounding the model in the spatial environment. In both Navigation and Verification, it is the first step to accurately identify the locations of the start, goal, and hazardous ice holes, converting the visual grid into a structured representation that can be used for planning.

- **DRAW2DPATH**
  - **Functionality:** A visualization and verification tool. It takes a starting coordinate and a sequence of directional commands (e.g., ['U', 'U', 'R']) and overlays the corresponding path onto the input image.

- **Role in VSP:** This tool externalizes the model's internal plan into a visual artifact. In Verification, it renders the given path for the model to judge. In Navigation, it serves as a final check, allowing the model to visually confirm that its generated path is correct and safe before outputting the final answer.

- **ASTAR**

  - **Functionality:** A classic planning algorithm encapsulated as a tool. It computes the shortest obstacle-free path between a start and a goal coordinate, given the locations of obstacles.

  - **Role in VSP:** This tool offloads the complex pathfinding computation. After the POINT tool identifies all key locations, ASTAR can be invoked to generate an optimal, logically sound path, freeing the MLLM to focus on higher-level task management and verification.

- **DETECTBLACKAREA**

  - **Functionality:** A specialized perception tool for the Jigsaw task. It analyzes an image and returns the bounding box coordinates of any completely black, rectangular regions, which correspond to the missing puzzle piece.

  - **Role in Jigsaw:** This tool provides a deterministic way to identify the "problem space". It is the critical first step in the solution trajectory, telling the model precisely where the candidate patches need to be inserted.

- **INSERTIMAGE**

  - **Functionality:** A visual manipulation tool. It takes a base image, a patch image, and a set of coordinates, and returns a new image where the patch has been inserted at the specified location.

  - **Role in Jigsaw:** This tool enables iterative hypothesis testing. The model uses it to physically place each candidate patch into the missing slot identified by DETECTBLACKAREA. The resulting composite image is then fed back to the model, allowing it to visually assess the quality of the fit.

- **CROP**

  - **Functionality:** An attentional tool. It takes an image and bounding box coordinates and returns a new, smaller image containing only the specified region.

  - **Role in GuiQA:** This tool mimics the human ability to focus on a specific part of a dense interface. By cropping a region of interest (e.g., a button or a text block), the model can reduce noise and ambiguity, creating a cleaner input for subsequent analysis by other tools or by the model itself.

- **OCR**

  - **Functionality:** An information extraction tool. It performs Optical Character Recognition on an input image, returning all detected text along with its corresponding bounding box coordinates.

  - **Role in GUIQA:** This tool grounds the model's understanding in the literal text present in the GUI. It is often used synergistically with CROP. After isolating a key UI element, the model can use OCR to reliably read its label (e.g., "Buy Now"), preventing the visual-only hallucinations common in MLLMs.

### B.2 HIGH-QUALITY COLD START TRAJECTORY DATA CURATION

For our structured reasoning tasks, we developed data generation and trajectory creation pipelines to ensure high quality and diversity. Some detailed data samples are shown in appendix B.3

**VSP** The VSP benchmark environments were procedurally generated using the **Gymnasium** (Towers et al., 2024) framework. To ensure variety, we systematically controlled the distribution of start points, end points, and ice holes. We synthesized environments of sizes $4 \times 4$, $6 \times 6$, and $8 \times 8$ for the training set, while reserving larger $5 \times 5$, $7 \times 7$, and $9 \times 9$ grids for testing. The Tool Cold Start (TC, SFT) trajectories were designed to mimic an optimal problem-solving process. For Navigation tasks, the trajectory consists of: (1) invoking the POINT tool to localize the start,

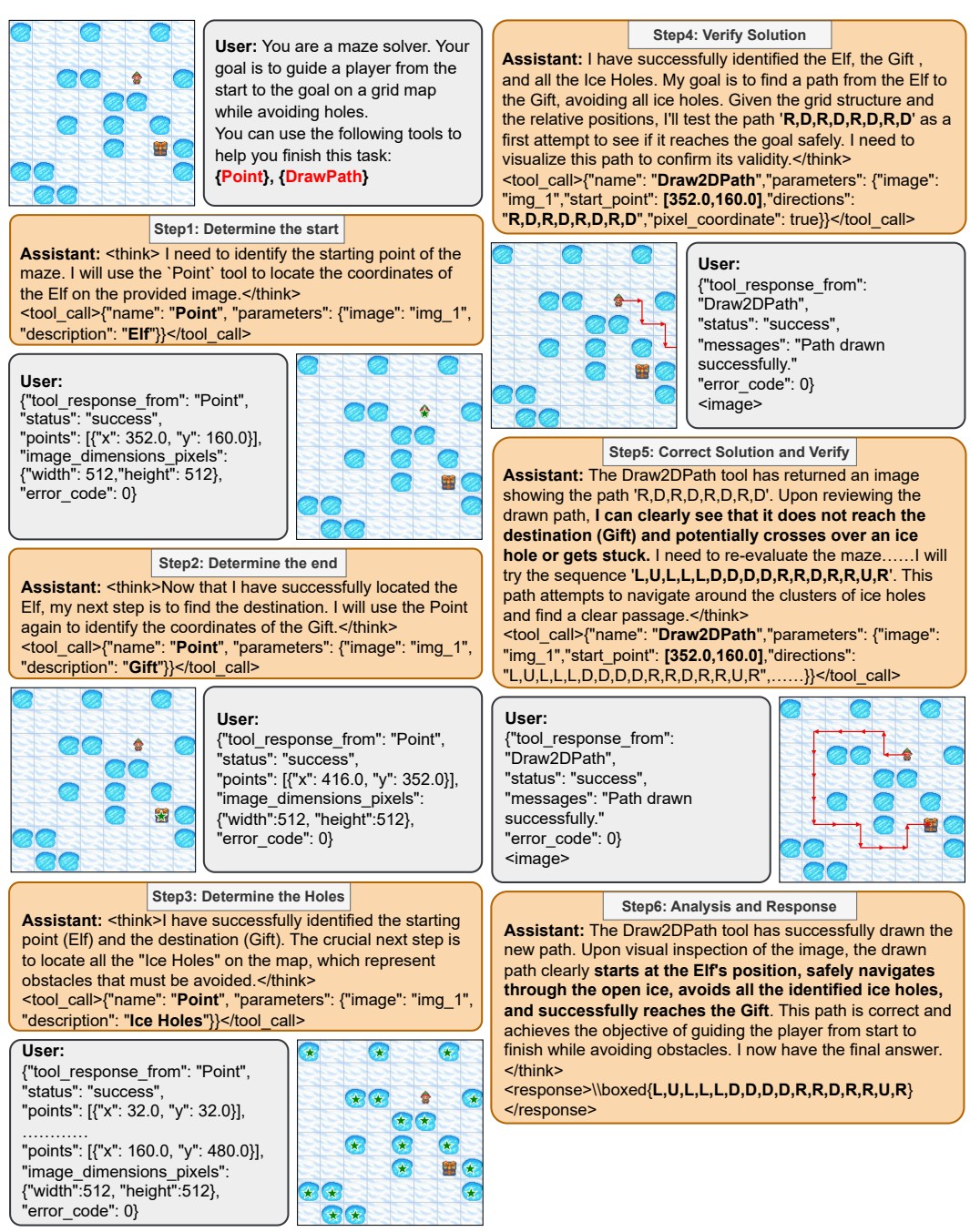

**Figure 6:** An example of a multi-turn cold-start trajectory for the VSP task.

end, and all ice holes; (2) performing textual reasoning based on these coordinates; and (3) calling DRAW2DPATH for final verification. Crucially, we also incorporated reflection and backtracking data derived from failure cases. For Verification tasks, the trajectory involves: (1) using POINT to locate the start, (2) employing DRAW2DPATH to render the proposed path, and (3) prompting the model to judge if this path intersects any ice holes.

**Jigsaw**   The Jigsaw dataset was constructed using images from the **COCO-2017** (Lin et al., 2014) training set. Each instance was created by first dividing an image into a $3 \times 3$ grid. A $2 \times 2$ sub-grid was then selected as the base image, from which one patch (e.g., top-right) was removed to create the problem. The removed patch served as the correct answer, while one of the remaining five patches from the original $3 \times 3$ grid was chosen as a distractor. The TC trajectory instructs the model to: (1)

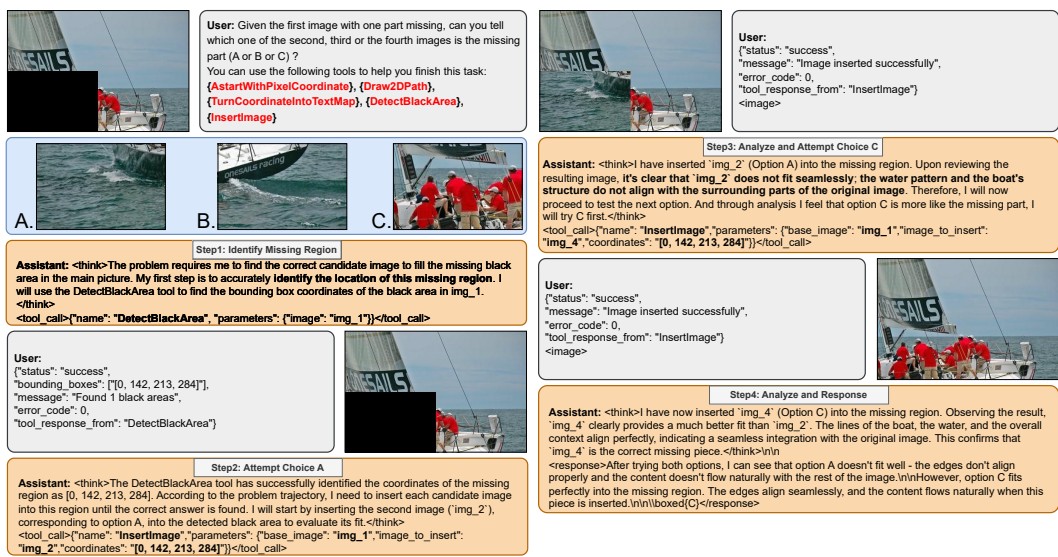

**Figure 7:** An example of a multi-turn cold-start trajectory for the Jigsaw task. This trajectory showcases an iterative trial-and-error process. The agent first uses DETECTBLACKAREA to identify the missing region. It then sequentially attempts to InsertImage with each candidate patch, analyzing the visual result of each attempt before arriving at the correct solution.

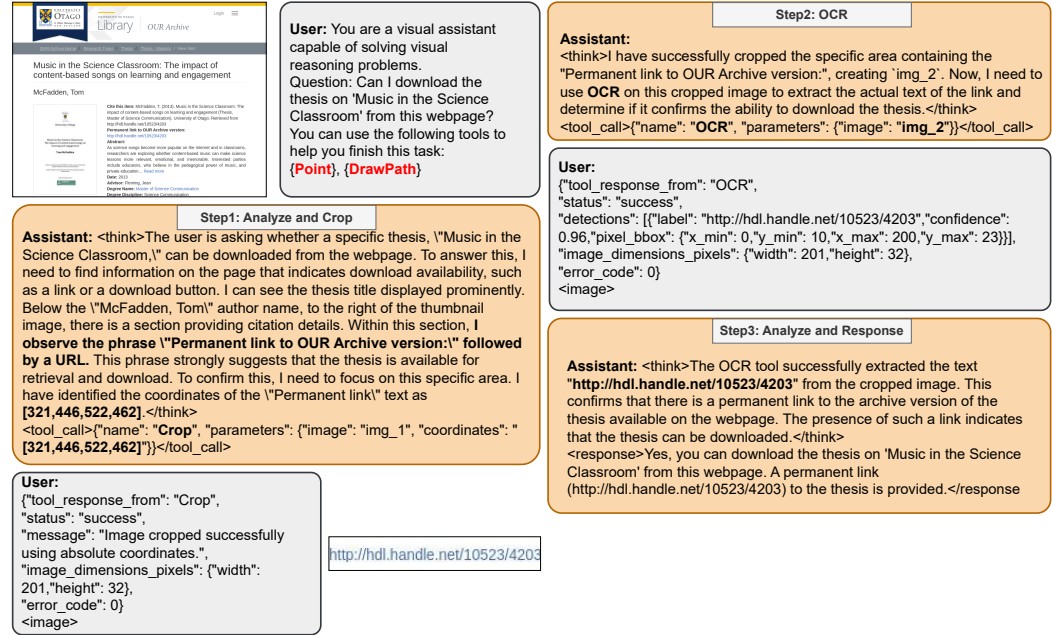

**Figure 8:** An example of a multi-turn cold-start trajectory for a GUI-QA task. This sample illustrates a focus-then-extract strategy. The agent first uses the CROP tool to isolate a specific, relevant section of the webpage. It then applies the OCR tool to this cropped, unambiguous input to perform precise information extraction.

call DETECTBLACKAREA to identify the coordinates of the missing section, and (2) iteratively call INSERTIMAGE for each candidate patch until the puzzle is solved. To enhance robustness and diversity, we introduced several key variations: ❶ The order of patch insertion attempts was randomized to ensure a uniform distribution of options. ❷ Scenarios involving tool failures (e.g., detection errors) were included, prompting the model to fall back on its intrinsic knowledge after several failed

attempts. ❸ A proportion of samples were designed to be solvable directly by the model without tool use, encouraging adaptive tool invocation.

**GUIQA** The process begins with 44k single-turn instances from the **Guichat** dataset. (Chen et al., 2024) To identify challenging cases that necessitate tool use, we first prompted a powerful vision-language model, Qwen-VL-2.5-72B (Bai et al., 2025), to answer the questions. We retained only the instances where the model failed, resulting in a subset of 7,100 "hard" questions. Next, for these 7,100 instances, we rendered the ground-truth answer coordinates as bounding boxes on the images. We then performed a manual visual inspection to ensure these boxes contained meaningful and relevant information, which filtered the set down to 1,800 valid data points. To generate high-fidelity tool-use trajectories for these cases, we provided the ground-truth answer and coordinates to gemini-2.5-flash Comanici et al. (2025), prompting it to produce the chain-of-thought reasoning and tool invocation sequence required to solve the problem. Finally, all generated trajectories were validated against our predefined format, and only those that strictly conformed were retained. This final curation step yielded a high-quality dataset of 1,139 instances for our TC stage.

After defining the abstract trajectory structure for all tasks, we followed a unified, two-stage process to create the final training data. First, we executed these trajectories programmatically to populate them with real tool inputs and outputs. Subsequently, we leveraged **Gemini 2.5 Flash** (Comanici et al., 2025) to generate the corresponding chain-of-thought (CoT) reasoning for each step. This process resulted in a final dataset of high-fidelity, tool-augmented trajectories complete with explicit reasoning chains, ready for our cold-start training.

### B.3 DATA SAMPLES

To provide a more concrete understanding of our cold-start data, we present representative multi-turn trajectory samples for each of our core tasks in Figures 6, 7, and 8. These examples are designed to showcase the sophisticated, human-like reasoning patterns we aim to instill in the model during the supervised fine-tuning phase.

The VSP sample (Figure 6) illustrates a methodical, multi-stage problem-solving process that includes perception, verification, and analysis. The Jigsaw sample (Figure 7) demonstrates an iterative trial-and-error strategy, where the agent actively evaluates the outcome of each tool call. Finally, the GUIQA sample (Figure 8) highlights a synergistic tool-use pattern, where one tool ('Crop') is used to create optimal conditions for another ('OCR'). Across all examples, the interplay between the model's internal thoughts ('<think>'), tool calls, and observations from the environment is clearly demonstrated.

### B.4 TOOL GRPO

Group Relative Policy Optimization (GRPO) is a reinforcement learning algorithm that evaluates policy performance by directly comparing a group of candidate reasoning trajectories. The process of Tool GRPO in AdaReasoner begins with the initial state $s_0 = \langle g, \text{text}, \text{image} \rangle$, for which the policy $\pi_\theta$ samples a set of $N$ complete trajectories as candidate responses, $\{\tau^1, \tau^2, \ldots, \tau^N\}$. Each trajectory is evaluated by a reward function, yielding rewards $r^i = R(\tau^i)$. GRPO then calculates a group-relative advantage $A^i$ for each trajectory by normalizing its reward against the statistics of the entire group:

$$A^i = \frac{r_i - \text{mean}\{r_1, r_2, \ldots, r_N\}}{\text{std}\{r_1, r_2, \ldots, r_N\}}.$$ 

(3)

The policy is then updated to favor trajectories with higher relative advantages by maximizing a clipped surrogate objective function. This objective is designed to ensure stable training by preventing excessively large policy updates. The full objective is:

$$J_{\text{GRPO}}(\theta) = \mathbb{E}_{q \sim P(Q), \{\tau^i\}_{i=1}^G \sim \pi_{\theta_{\text{old}}}(\cdot|q)}$$
$$\left[ \sum_{i=1}^{G} \sum_{j=1}^{|\tau^i|} \frac{1}{G|\tau^i|} \min\left( m_j^i A_i, \text{clip}\left( s_j^i, 1 - \varepsilon, 1 + \varepsilon \right) A_i \right) - \beta \, \mathbb{D}_{\text{KL}}\left( \pi_\theta \, \| \, \pi_{\text{ref}} \right) \right].$$

(4)

Here, $m_j^i = \frac{\pi_\theta(\tau_j^i - s_i|s_i)}{\pi_{\theta_{old}}(\tau_j^i - s_i|s_i)}$ is the importance sampling ratio that measures the change between the new policy $\pi_\theta$ and the old policy $\pi_{\theta_{old}}$ used to generate the samples. The Kullback-Leibler (KL) divergence penalty, $\mathbb{D}_{KL}(\pi_\theta||\pi_{\text{ref}})$ regularizes the policy update by penalizing large deviations from a reference policy $\pi_{\text{ref}}$.

**Reward Design**    Our reward function is designed to evaluate both the structural syntax and the semantic correctness of the model's output. The total reward, $R_{\text{total}}$, is a composite score defined as:

$$R_{\text{total}} = R_{\text{format}} \cdot (\lambda_{\text{tool}} \cdot R_{\text{tool}} + \lambda_{\text{acc}} \cdot R_{\text{acc}}) \tag{5}$$

Here, $R_{\text{format}}$ acts as a binary gate, ensuring that rewards for tool usage ($R_{\text{tool}}$) and final answer accuracy ($R_{\text{accuracy}}$) are granted only if the output adheres to the required structure. This design incentivizes the model to first master the correct syntax before optimizing for functional correctness. $\lambda_{\text{acc}}$ is a hyperparameter and the effect of it is discussed in Section 5.

- **Format Reward ($R_{\textbf{format}}$)** The format reward is a binary signal that assesses the structural integrity of the model's output. It verifies that the generated response contains all required special tokens in the correct order and follows predefined rules.

$$R_{\text{format}} = \begin{cases} 1 & \text{if the output format is valid} \\ 0 & \text{otherwise} \end{cases} \tag{6}$$

  If $R_{\text{format}}$ is 0, the total reward for the trajectory is nullified, creating a strong imperative for the model to learn the required output structure.

- **Tool Reward ($R_{\textbf{tool}}$)** The tool reward provides a fine-grained evaluation of the tool-calling process, with a score ranging from 0 to 4. We employ a hierarchical scoring system where each level must be passed to proceed to the next.

  1. **Invocation Structure (Score 1):** A score of 1 is awarded if the tool call is correctly encapsulated within the `<tool_call>` and `</tool_call>` tokens. If not, the score is 0, and no further tool evaluation occurs.
  2. **Tool Name Validity (Score 2):** If the structure is correct, we verify that the invoked tool name exists in the set of available tools, $\mathcal{T}$. If the name is valid, the score becomes 2.
  3. **Parameter Name Correctness (Score $\in [2, 3]$):** If the tool name is valid, we assess the parameter names. A partial score is awarded based on the proportion of correctly named parameters. A perfect match yields a score of 3. The score is calculated as:

$$R_{\text{tool}} = 2 + \frac{|\text{params}_{\text{correct\_name}}|}{|\text{params}_{\text{total}}|} \tag{7}$$

  4. **Parameter Content Validity (Score $\in [3, 4]$):** Finally, if all parameter names are correct (base score of 3), we evaluate the parameter values for semantic and contextual validity. The final score is proportional to the number of correct values, reaching a maximum of 4.

$$R_{\text{tool}} = 3 + \frac{|\text{params}_{\text{correct\_content}}|}{|\text{params}_{\text{total}}|} \tag{8}$$

- **Accuracy Reward ($R_{\textbf{accuracy}}$)** The accuracy reward evaluates the final outcome of the reasoning process, providing a clear signal based on the correctness of the model's final answer.

$$R_{\text{accuracy}} = \begin{cases} 4 & \text{if the final answer is correct} \\ 0 & \text{otherwise} \end{cases} \tag{9}$$

This multi-faceted reward structure guides the model toward not only achieving the correct final answer but also mastering the intermediate steps of correct formatting and precise tool invocation.

## C    EXPERIMENT DETAILS

### C.1    TASK DEFINITION

We evaluate our approach across a diverse suite of three challenging tasks to validate whether our approach can help enhance the reasoning ability.

**Visual Spatial Planning**    We adopt the FrozenLake scenario (Towers et al., 2024) to evaluate models' spatial planning and verification abilities. The **navigation** task requires the model to generate a safe path from the start to the goal while avoiding holes, which demands accurate perception of the grid map and robust sequential reasoning to plan multi-step trajectories. The **verification** task instead focuses on state checking, determining whether a given location or a proposed path is safe, which isolates the perception and reasoning components of the planning pipeline. Together, these tasks expose two fundamental challenges for VLMs: (i) precise visual perception of spatial layouts under varying map sizes, and (ii) reliable reasoning over action sequences to ensure safety and goal completion.

Concretely, we evaluate models on two benchmarks. The first is **VSPO**, a dataset we construct to assess out-of-distribution generalization. During training, we use maps of sizes $4 \times 4$, $6 \times 6$, and $8 \times 8$, while reserving maps of sizes $3 \times 3$, $5 \times 5$, $7 \times 7$, and $9 \times 9$ for testing. This setup not only probes the model's ability to generalize to unseen spatial configurations but also examines whether it truly learns to leverage tool usage for problem solving. The second is the original **VSP benchmark** (Wu et al., 2024), where we adopt the navigation and verification tasks to further test visual-spatial reasoning and state-checking capabilities under standardized settings.

**Jigsaw**    The Jigsaw task (Noroozi & Favaro, 2016) evaluates a model's ability to reconstruct holistic understanding from fragmented visual inputs. Specifically, the model must infer the correct spatial arrangement of shuffled image patches and reason about their part–whole relationships. This requires fine-grained perception to capture local details, as well as global reasoning to integrate them into a coherent whole. The key challenges lie in bridging local–global consistency and maintaining semantic alignment across patches, making it a strong test of visual compositionality and structural reasoning.

Concretely, we evaluate models on two benchmarks. The first is **Jigsaw-COCO**, where we construct training and test splits based on the COCO train 2017 dataset (Lin et al., 2014). We extract the top-left, top-right, and bottom-left patches of each image to form the training set, while reserving the bottom-right patches for testing. This design allows us to probe the model's out-of-distribution generalization and examine whether it truly learns to invoke tool usage for solving the puzzle. The second is the **Jigsaw benchmark from BLINK (BLINK-J)** (Fu et al., 2024), which provides a standardized evaluation of fine-grained visual reasoning and compositional understanding under more challenging and diverse settings.

**GUIQA**    The GUIQA task is designed to evaluate a model's sophisticated capabilities in fine-grained visual understanding and critical information extraction from GUIs. In this task, a model is provided with a GUI screenshot and an associated question, where the main difficulty lies in precisely grounding UI elements on a dense layout, comprehending their functional roles, and performing multi-step reasoning by integrating scattered information.

The evaluation is conducted on two distinct datasets. The first is the **GUIChat** (Chen et al., 2024), which specifically probes the model's capacity for interactive, dialogue-based comprehension of webpage screenshots. Models are required to process complex queries related to visual information, human-centric needs, world knowledge, and reasoning. The second is the **WebQA** from the **Web-MMU** (Awal et al., 2025). It offers a structured evaluation across three distinct categories. Agentic Action tests the ability to understand UI elements like buttons and menus in order to formulate the necessary user actions, complete with precise spatial grounding. General Visual Comprehension assesses how well the model can extract and synthesize information from varied page components, including text, images, and graphics. And Multi-step Reasoning demands complex inference, numerical calculations, and comparisons across different parts of the UI.

## C.2    PROMPTS

The system prompt used for guiding the tool-planning model is provided in Figure 9.

## C.3    IMPLEMENTATION DETAILS

We developed **Tool Factory**, an end-to-end framework that orchestrates the entire lifecycle of our tool-planning models, from data curation to evaluation. At the heart of this framework is the Tool

---

**System Tool Prompt**

You are a visual assistant capable of solving visual reasoning problems. You can rely on your own capabilities or use external tools to assist in solving.
Available Tools In your response, you can use the following tools:
{Tool List}
Steps for Each Turn
1. **Think:** First, silently analyze the user's request to understand the goal. This thinking process should be enclosed in `<think>` and `</think>` tags.
2. **Decide Action:** Based on your thinking, decide on one of the following two actions:
- **If you need to use a tool:** Generate your tool call, enclosed between `<tool_call>` and `</tool_call>` tags. **Do not** generate a `<response>` in this turn.
- **If you have enough information to answer:** Generate your final, user-facing answer, enclosed between `<response>` and `</response>` tags. **Do not** generate a `<tool_call>` in this turn.
Output Format:
Your output must always begin with your thought process. After the `<think>` block, you must provide either a `<tool_call>` or a `<response>`, but **never both** in the same turn.
**Case 1: Tool Use is Required**
`<think>` Your thoughts and reasoning `</think>`
`<tool_call>`
{"name": "Tool name", "parameters": {"Parameter name": "Parameter content", "...": "..."}}
`</tool_call>`
**Case 2: Ready to Respond to the User**
`<think>` Your thoughts and reasoning `</think>`
`<response>` Your final response `</response>`
Important Notes
1. You must always include the `<think>` field to outline your reasoning. Provide one of `<tool_call>` or `<response>`. You must not include both `<tool_call>` and `<response>` in the same turn because they are mutually exclusive. If tool usage is required, you must instead include both `<think>` and `<tool_call>`, and omit `<response>` for that turn. If no further tool usage is required and ready to answer the user's question, you can then use `<think>` to summarize your reasoning and include `<response>` with your final answer, and this indicates the ends of the conversation.
2. You can only invoke a single tool call at a time in the `<tool_call>` fields. The tool call should be a JSON object with a "name" field and a "parameters" field containing a dictionary of parameters. If no parameters are needed, leave the "parameters" field an empty dictionary. All images have their coordinate origin at the top-left corner.
3. Some tools require image input. You do not need to generate or upload the actual image data simply refer to an image using a placeholder in the form of "img_n". There may be multiple images present in the dialogue. Besides the original image, additional images may appear as a result of prior tool calls (e.g., edited images returned by visual editing tools). You are free to select which image to use as input for the next tool. The index n in "img_n" refers to the image's position in the dialogue history:
- The original image is always referred to as "img_1".
- Each subsequent image, including those returned from tools, is assigned "img_2", "img_3", and so on, in the order they appear in the dialogue.
For example:{"parameters": {"image": "img_1", "other_params": "other_values"}}
4. All image coordinates used must be in absolute pixel values, not relative or normalized coordinates.
5. At the end, provide your final answer by placing it inside boxed{}, and wrap the entire final output inside `<response></response>` tags.

**Figure 9:** Our system employs tool prompts to guide models in learning how to use tools effectively.

**Table 7:** Tool Cold Start (SFT) Training Configuration and Hyperparameters.

| Category | Hyperparameter | Value / Setting |
|---|---|---|
| **Model** | Base Model | Qwen2.5-VL-7B-Instruct |
| | Vision Tower Frozen | True |
| | MM Projector Frozen | True |
| | Finetuning Type | Full |
| | DeepSpeed Stage | ZeRO-3 |
| **Dataset** | Max Samples | 332,649 |
| | Cutoff Length | 35,536 |
| | Preprocessing Workers | 64 |
| **Training** | Batch Size per Device | 1 |
| | Gradient Accumulation Steps | 2 |
| | Effective Batch Size | 2 |
| | Learning Rate | 1e-5 |
| | Epochs | 3 |
| | LR Scheduler | cosine |
| | Warmup Ratio | 0.1 |
| | Mixed Precision | bfloat16 |
| **Logging / IO** | Logging Steps | 10 |
| | Checkpoint Save Steps | 100 |
| **Evaluation** | Train/Validation Split | 90% / 10% |
| | Eval Batch Size per Device | 1 |
| | Eval Steps | 100 |

Server, a unified, MCP-like service that manages all available tools, from simple offline utilities to compute-heavy online expert models.

### C.3.1 DATA CURATION

During the data curation stage, we employ our Tool Curation Module, which leverages the Tool Server to automatically generate high-quality cold-start trajectories. Specifically, we first design abstract, optimal problem-solving blueprints for each task, consisting of a tool-call chain and chain-of-thought (CoT) placeholders. We then prompt Gemini-2.5-Flash to fill these placeholders with detailed CoT reasoning. Finally, the Tool Server executes the corresponding tool calls and integrates the results into the dialogue, yielding a complete and coherent training instance.

### C.3.2 TOOL COLD START STAGE

During the cold-start stage, these trajectories are used for full-parameter supervised fine-tuning, for which we adopt the LLaMA Factory framework (Zheng et al., 2024). The key configurations and hyperparameters are summarized in Table 7.

### C.3.3 TOOL GRPO STAGE

Following SFT, the model is further refined in the Tool GRPO stage using ToolRL, our custom reinforcement learning framework inspired by Sheng et al. (2024); Zheng et al. (2025), which also relies on the Tool Server for live tool interactions. Finally, for systematic performance assessment, we developed TF-Eval, our dedicated evaluation framework. TF-Eval interacts with the Tool Server to benchmark our tool-planning models across a diverse suite of multimodal tasks. The key configurations and hyperparameters of Tool GRPO stage are summarized in Table 8.

## D DISCUSSION

A central finding of our work concerns the dual role of the `Tool-Cold-Start` (SFT) phase, which highlights a critical trade-off between imparting expert knowledge and preserving a model's

**Table 8:** Key configurations and hyperparameters used in the Tool GRPO stage.

| Category | Hyperparameter | Value / Setting |
|---|---|---|
| **Data** | Max Prompt Length | 8192 tokens |
| | Max Response Length | 20480 tokens |
| | Train Batch Size | 32 |
| | Shuffle | True |
| | Filter Overlong Prompts | True |
| **Policy** | Strategy | FSDP |
| | Gradient Checkpointing | True |
| | PPO Mini-batch Size | 8 |
| | PPO Micro-batch Size / GPU | 1 |
| | Max Token Len / GPU (PPO) | 16384 |
| | Grad Clip | 1.0 |
| | Clip Ratio (PPO) | 0.2 |
| | PPO Epochs | 1 |
| | Entropy Coeff | 0.0 |
| | Use KL Loss | False |
| | Actor LR | 1e-6 |
| | Weight Decay | 0.01 |
| | FSDP Param Offload | True |
| | FSDP Optimizer Offload | True |
| | # Nodes / GPUs | 1 node, 8 GPUs |
| **Rollout** | Engine | vLLM |
| | Temperature | 1.0 |
| | Top-p | 1.0 |
| | Top-k | -1 |
| | # Samples per Prompt (n) | 4 |
| | Dtype | bfloat16 |
| | Tensor Model Parallel Size | 2 |
| | Max # Batched Tokens | 32768 |
| | GPU Memory Utilization | 0.65 |
| | Enforce Eager | False |
| | Chunked Prefill | False |
| **Tool-Agent** | Max Turns per Episode | 10 |
| **Critic** | Strategy | FSDP |
| | LR | 1e-5 |
| | Weight Decay | 0.01 |
| | PPO Epochs | 1 |
| | Grad Clip | 1.0 |
| **Algorithm** | Advantage Estimator | GRPO |
| | Gamma | 1.0 |
| | Lambda | 1.0 |
| | Use KL in Reward | False |
| | KL Coef | 0.0 |
| | Norm Adv by Std in GRPO | True |

exploratory freedom. Our results suggest that the decision to include a supervised pre-training stage is not universally beneficial, but rather highly contingent on the nature of the task.

For complex, structured tasks with discernible optimal solutions, such as VSP and Jigsaw, the SFT phase provides a decisive advantage. In these scenarios, discovering an effective tool-use trajectory from scratch is a non-trivial exploration problem for the model due to its inherent reasoning or knowledge deficits. By exposing the model to high-quality, deterministic solution paths, the `Tool-Cold-Start` phase effectively bootstraps the learning process, instilling a strong inductive bias towards a correct strategy. The empirical results in Table 2 validate this unequivocally: for our 7B models, adding this SFT phase before Tool-GRPO yields a massive performance improvement of **+24.93** points on VSP and **+19.82** points on Jigsaw compared to using Tool-GRPO alone.

Conversely, for open-ended and highly generalized domains like GUIQA, the limitations of this pre-defined guidance become apparent. In such settings, the optimal tool-use strategy is often unknown even to human designers, making any human-designed trajectory likely sub-optimal. We find that a rigid SFT phase can inadvertently restrict the model's exploratory freedom during subsequent RL by creating a strong policy bias, which hinders the discovery of more effective, emergent strategies. This effect is clearly observed in our results for the 7B model on the WebMMU benchmark, where the standalone Tool-GRPO approach actually outperforms the combined pipeline (**72.97 vs. 68.16**).

This dichotomy suggests a key principle for training tool-augmented agents: while injecting expert knowledge via SFT is a powerful method for tasks with well-defined solution spaces, a pure reinforcement learning approach like Tool-GRPO may be superior for more dynamic and general tasks that benefit from unconstrained exploration.

