# OpenReview forum: "AdaReasoner: Dynamic Tool Orchestration for Iterative Visual Reasoning"
_ICLR.cc/2026/Conference — ICLR 2026 Poster_

### Official Review · Reviewer_iRNU · 2025-10-27

**Soundness:** 4
**Presentation:** 3
**Contribution:** 3
**Rating:** 8
**Confidence:** 3

**Summary:**

The paper “AdaReasoner: Dynamic Tool Orchestration for Iterative Visual Reasoning” presents a new framework for enhancing multimodal large language models (MLLMs) with adaptive, multi-turn tool-use capabilities. Existing methods generally employ fixed, single-tool reasoning or scripted tool invocation, limiting their capacity for dynamic, multi-step planning. To address this, the authors introduce AdaReasoner, a system that integrates a diverse suite of visual tools and introduces a specialized Tool GRPO reinforcement learning algorithm alongside a Tool Cold Start phase for data-driven initialization. The system uses a curated dataset of reasoning trajectories that include reflection, backtracking, and tool failure cases—teaching models to plan adaptively and recover from errors. Extensive experiments demonstrate that AdaReasoner significantly improves reasoning performance, achieving +38.7% average accuracy gains on 7B models and near-perfect accuracy (97.6%) on Visual Spatial Planning tasks—outperforming both open and proprietary systems like GPT-5 and Claude Sonnet 4. The framework also exhibits emergent behaviors, such as autonomously adopting or discarding tools depending on their utility.

**Strengths:**

The paper is well-motivated and grounded in both conceptual and empirical depth. It identifies a clear and timely gap in MLLM research—static or single-tool reasoning—and develops a complete system that addresses it with a mix of structured methodology and adaptive reinforcement learning. The data curation pipeline (with reflection and failure handling) is a major strength, as it systematically teaches models resilience and iterative self-correction. The Tool GRPO formulation introduces a carefully designed reward structure that balances correctness, formatting, and tool-usage efficiency, enabling stable optimization of multi-turn trajectories. The ablation studies and visualization of tool-call frequency provide valuable insight into the model’s learning dynamics, confirming the emergence of adaptive behaviors such as adopting useful tools (A*) or discarding irrelevant ones. Furthermore, the analysis connecting perception tools (“see”), manipulation tools (“verify”), and planning tools (“calculate”) adds interpretability and conceptual clarity. Overall, the work is good, empirically convincing, and establishes AdaReasoner as a strong contribution to multimodal tool-use research.

**Weaknesses:**

the paper leaves several conceptual and methodological aspects underexplored that could make the study more complete. Although the reward function and reflection-based trajectories are described, their quantitative sensitivity (e.g., how different reward weights or reflection ratios affect learning) is not studied, limiting reproducibility insights. While performance improvements are substantial, the evaluation scope is confined to visual reasoning tasks, and cross-modal generalization (e.g., audio or temporal modalities) is not examined, despite the framework’s claimed generality.

**Questions:**

please check above

---

> ### Author Response · Authors · 2025-11-21
> **Rebuttal for Reviewer iRNU (Part 1)**
>
> We would like to thank the reviewer for your insightful comments and constructive suggestions.
>
> **W1: Although the reward function and reflection-based trajectories are described, their quantitative sensitivity (e.g., how different reward weights or reflection ratios affect learning) is not studied, limiting reproducibility insights.**
>
> **WR1:** To address your concern regarding the quantitative sensitivity of our reward design, we conduct an ablation study on the VSP task by varying the reward-weight configurations used during RL. Our reward function can be formalized as:
>
> $R_{\text{total}} = R_{\text{format}} \cdot (\lambda_{\text{tool}} \cdot R_{\text{tool}} + \lambda_{\text{acc}} \cdot R_{\text{acc}})$
>
> We systematically adjust $\lambda_{\text{tool}}$ and $\lambda_{\text{acc}}$ to evaluate their influence on learning dynamics and final performance. Specifically, we train the model on the same VSP task data for 100 RL steps under different reward-weight settings, monitor the training curves to ensure convergence, and then evaluate each checkpoint's performance. The results are summarized in the table below.
>
>
> |   $\lambda_{\text{tool}} : \lambda_{\text{acc}}$ | VSP Nav (%)   | VSP Verify (%)   | VSP Overall (%)   | VSP-test Nav (%)   | VSP-test Verify (%)   | VSP-test Overall (%)   |
> |------------:|:--------------|:-----------------|:------------------|:-------------------|:----------------------|:-----------------------|
> |         0:1 | 51.83         | 95.00            | 71.45             | 41.78              | 75.58                 | 57.37                  |
> |         1:2 | 49.50         | 95.80            | 70.55             | 36.44              | 94.29                 | 63.11                  |
> |         1:1 | 64.00         | 96.40            | 78.73             | 48.56              | 96.23                 | 70.54                  |
> |         2:1 | **90.33**     | **96.80**        | **93.27**         | **70.33**          | **96.36**             | **82.34**              |
>
> As shown in the table, the model's performance consistently improves as the ratio $\lambda_{\text{tool}} : \lambda_{\text{acc}}$ increases. This indicates that larger tool rewards not only accelerate convergence during RL training but also lead to significantly better final performance. These results validate that our tool-reward design is effective and plays a crucial role in helping the model learn tool calling more efficiently and robustly.

---

> ### Author Response · Authors · 2025-11-21
> **Rebuttal for Reviewer iRNU (Part 2)**
>
> **W2: Evaluation Lacks Cross-Modal Generalization Beyond Visual Reasoning Tasks**
>
> **WR2:** We address this question by evaluating the generalizability of our method on several widely used, general-purpose benchmarks. The model we use is an MLLM designed for multimodal understanding, which is not capable of cross-modal generalization (e.g., to audio or temporal signals). However, to assess the generality of our framework beyond structured visual reasoning, we additionally evaluate it on several more general multimodal understanding tasks, including WebQA, GUI-based reasoning, and real-world Visual Search.
>
> Concretely, we first merge the Tool Cold Start data from the three tasks and randomize the tool names, parameters, and descriptions. We then curate a broader RL dataset that includes Visual Reasoning, Jigsaw, WebQA, and Visual Search data. After applying Tool GRPO across these four tasks, we obtain our final model. We evaluate this model on several widely used benchmarks, and the results are presented in the table below.
>
> | Model | VSPO Nav | VSPO Verify | VSPO Overall | VSP Nav | VSP Verify | VSP Overall | Jigsaw | BLINK-J | GUIChat | WebMMU Act | HRBench [1] | V* [2] |
> |-------|-----------|--------------|----------------|----------|-------------|--------------|---------|-----------|-----------|-------------|----------|-------|
> | GPT 5 | 26.89 | 42.86 | 34.25 | 48.17 | 64.60 | 55.64 | 80.10 | 73.33 | 71.41 | 80.49 | 74.38 | 74.87 |
> | Claude 4 sonnet | 37.56 | 67.92 | 51.56 | 48.17 | 66.00 | 56.27 | 58.60 | 65.33 | **93.14** | 83.54 | 60.62 | 59.69 |
> | Qwen 2.5 VL 7B | 5.22 | 48.96 | 25.39 | 12.33 | 47.00 | 28.09 | 45.70 | 52.67 | 68.09 | 67.48 | 63.62 | 63.35 |
> | Qwen 2.5 VL 32B | 7.56 | 53.12 | 28.56 | 24.33 | 45.40 | 33.91 | 59.50 | 64.67 | 85.21 | 85.98 | 70.12 | 72.25 |
> | Qwen 2.5 VL 72B | 17.22 | 52.34 | 33.41 | 28.00 | 52.40 | 39.09 | 70.10 | 71.33 | _88.01_ | _91.06_ | 73.00 | _80.10_ |
> | InternVL3 78B | 7.22 | 52.60 | 28.14 | 21.67 | 51.20 | 35.09 | 52.80 | 60.00 | 79.83 | 71.34 | _75.12_ | **81.15** |
> | Qwen 2.5 VL 7B (Baseline) | 5.22 | 48.96 | 25.39 | 12.33 | 47.00 | 28.09 | 45.70 | 52.67 | 68.09 | 67.48 | 63.62 | 63.35 |
> | **Ours** | **72.33** | **95.32** | **82.93** | **91.50** | **95.40** | **93.27** | **94.10** | **93.33** | 80.15 | 77.03 | 69.88 | 75.92 |
> | Δ | +67.11 | +46.36 | +57.54 | +79.17 | +48.40 | +65.18 | +48.40 | +40.67 | +12.06 | +9.55 | +6.26 | +12.57 |
>
> As shown in the table, our model achieves clear gains on both WebQA-style and real-world visual search tasks, including +12.06 on GUIChat, +6.26 on HRBench, and +12.57 on V*. These results demonstrate that our method not only substantially enhances the model's visual reasoning ability but also improves its open-ended reasoning capability through general-purpose tools such as OCR and Crop, even without relying on task-specific tools.
>
> Moreover, we agree that the cross-modal generalization scenarios you mentioned represent an important and valuable direction for tool-augmented models which is still under-explored. Such tasks offer promising opportunities for extending tool-based augmentation beyond vision-centered tasks. We view this as an exciting avenue for future research and a natural next step for enhancing the generality of tool-augmented MLLMs.
>
> **Reference**
>
> [1] Wang, Wenbin, et al. "Divide, conquer and combine: A training-free framework for high-resolution image perception in multimodal large language models." Proceedings of the AAAI Conference on Artificial Intelligence. Vol. 39. No. 8. 2025.
>
> [2] Wu, Penghao, and Saining Xie. "V*: Guided visual search as a core mechanism in multimodal llms, 2023." URL https://arxiv.org/abs/2312.14135 5.

---

> ### Author Response · Authors · 2025-11-27
> **Thank you & Looking Forward to Further Discussion**
>
> Dear Reviewer iRNU,
>
> Thank you once again for your insightful comments, which have helped us improve the rigor and scope of our work. As the discussion period continues, we wish to briefly summarize how our rebuttal and new experiments address your main concerns:
> * **Quantitative Sensitivity of Reward Function (W1):** To address your concern about reproducibility, we conducted a new ablation study on the reward weights. The results clearly show that placing a higher weight on the tool-correctness reward ($\lambda_{\text{tool}}$) consistently and significantly improves performance, validating our reward design and its crucial role in the learning process.
> * **Generalization Beyond Visual Reasoning (W2):** While our model is vision-language specific, we tested its generalization on a range of diverse, open-ended multimodal benchmarks, including WebQA (GUIChat) and Visual Search (HRBench, V*). Our method achieved significant performance gains on these tasks, demonstrating that the learned capabilities are not confined to structured visual reasoning but are broadly applicable.
>
>
> We hope these clarifications and new experimental results have fully addressed your concerns. Your feedback has been invaluable, and we are happy to engage in any further discussion.
>
> Warm regards,
>
> The Authors

---

### Official Review · Reviewer_W5WP · 2025-10-31

**Soundness:** 3
**Presentation:** 4
**Contribution:** 3
**Rating:** 8
**Confidence:** 4

**Summary:**

This paper introduces AdaReasoner, a framework for dynamic tool orchestration in MLLMs. Unlike prior approaches that rely on fixed, single-step tools, AdaReasoner allows models to plan, select, and compose tools adaptively across multiple reasoning turns. The framework combines: 1. a data curation pipeline that produces multi-turn tool-use trajectories with reflection and backtracking, and 2. a Tool-GRPO reinforcement learning algorithm designed to optimize these multi-turn trajectories with adaptive rewards.
Empirically, AdaReasoner significantly boosts reasoning accuracy of open-source vision–language models (e.g., Qwen2.5-VL-7B) across tasks like Visual Spatial Planning, Jigsaw, and GUI-QA, achieving up to +38.7% average improvement and even outperforming large proprietary models such as GPT-5 and Claude Sonnet 4. The authors also report emergent adaptive behaviors, where the model autonomously learns to adopt or discard tools depending on task context.

**Strengths:**

The authors demonstrate on their method improves over a variety of baselines across several visual reasoning benchmarks, with sufficient ablation experiments as well. A common limitation about training-based approaches for tool integrated reasoning is that they may not generalize to introduced tools. The authors address this by showing that at inference time, adding an unseen tool (A*) improves performance.

**Weaknesses:**

The authors show that RL training allows the model to learn how much to use different tools ("adopt", "discard", "modulate") and call this an emergent behavior at multiple points throughout the paper. However, the way the authors use the term "emergent behavior" could benefit from some clarification / definition. Generally, emergent behaviors refer to nonobvious / surprising capabilities not explicitly optimized in the object and generally only "emerge" at scale. In this case, the method is deliberately training the model to perform tool calls (with cold start and GRPO). Maybe learning adaptive tool selection, tool use optimization, etc are more precise terms to be used in this setting.

**Questions:**

For proprietary model baselines, could the authors give more specific details on they were run? Were they prompted to answer in one turn? If the proprietary models were evaluated using frameworks that involve explicit multistep planning and reasoning, e.g. ReAct (https://arxiv.org/abs/2210.03629) or OctoTools (https://arxiv.org/abs/2502.11271), how would their performance compare to TC and TG methods?
Could the authors clarify on how they are using the term "emergent" behavior, strategies, etc (see Weaknesses)?
Are there ablation experiments varying the amount of compute / date used in cold-start affect downstream performance (with and without RL)?
Could the authors include more details on training, e.g. hyperparameters?
Will the cold-start trajectory dataset be released upon acceptance?
Minor: figure 2 bottom panel is labeled (c) instead of (b)

---

> ### Author Response · Authors · 2025-11-21
> **Rebuttal for Reviewer W5WP (Part 1)**
>
> We are grateful to the reviewer for the constructive feedback and insightful guidance.
>
> **Q1: For proprietary model baselines, could the authors give more specific details on how they were run?**
>
> **A1:** Thank you for this valuable suggestion. While Section 4.1 briefly describes how we evaluate the proprietary model baselines, we acknowledge that the current level of detail may be insufficient. Below, we provide a more complete description of the evaluation setup, and we will revise the manuscript to include these details in the main text or in the appendix to ensure full transparency and reproducibility.
>
> |Item|Description|
> |-------|----------|
> |GPT5 Version|GPT-5-20250807|
> |Claude Version|Claude-sonnet-4-20250514|
> |Gemini Version|Gemini-2.5-flash (accessed on Sep 14, 2025)|
> |Temperature |0|
> |Max New Tokens |2048|
> |System Prompt|You are a visual assistant capable of solving visual reasoning problems. Provide your final answer by placing it inside \boxed{}|
> |ICL setting|N/A|
> |Qustion Rounds|1|
> ****
> **Q2: Were they prompted to answer in one turn?**
>
> **A2:** Yes, they are prompted to answer the question in one turn without in-context learning settings.
> ****
> **Q3: If the proprietary models were evaluated using frameworks that involve explicit multistep planning and reasoning, e.g., ReAct  or OctoTools, how would their performance compare to TC and TG methods?**
>
> **A3:** To address this question, we re-evaluate both proprietary and strong open-source models under a tool-enhanced setting, so that their performance can be fairly compared against our TC and TG methods.
>
> Our tool-server framework itself can function as a general tool-enhancement framework, similar in spirit to ReAct or OctoTools, and we adopt it for all models to ensure consistency and ease of implementation. Concretely, we select GPT-5, Qwen2.5-VL-7B, and Qwen2.5-VL-72B, and evaluate each model with and without tool-enhancement enabled. We then compare their performance against our TC+TG method. The results are summarized in the table below.
>
> | **Model** | **VSPO** |  |  | **VSP** |  |  | **Jigsaw** | **BLINK-J** | **GUIChat** | **WebMMU** |  |  |  |
> | --- | --- | --- | --- | --- | --- | --- | --- | --- | --- | --- | --- | --- | --- |
> |  | Nav | Verify | Overall | Nav | Verify | Overall |  |  |  | Avg. | Act. | Compre. | Rea. |
> | GPT 5 | 26.89 | 42.86 | 34.25 | 48.17 | 64.60 | 55.64 | 80.10 | 73.33 | 71.41 | 62.13 | 80.49 | 68.65 | 45.96 |
> | GPT 5 + Tools | *38.11* | *69.87* | *52.75* | *61.67* | *83.00* | *71.36* | *84.50* | *76.00* | 76.51 | **77.44** | *88.82* | **78.88** | **68.58** |
> | Qwen 2.5 VL 7B | 5.22 | 48.96 | 25.39 | 12.33 | 47.00 | 28.09 | 45.70 | 52.67 | 68.09 | 59.08 | 67.48 | 69.31 | 48.46 |
> | Qwen 2.5 VL 7B + Tools | 11.89 | 48.96 | 28.98 | 17.83 | 45.60 | 30.45 | 45.00 | 56.00 | 56.76 | 54.95 | 69.51 | 63.04 | 40.82 |
> | Qwen 2.5 VL 72B | 17.22 | 52.34 | 33.41 | 28.00 | 52.40 | 39.09 | 70.10 | 71.33 | *88.01* | *77.10* | **91.06** | *74.59* | *68.14* |
> | Qwen 2.5 VL 72B + Tools | 29.56 | 51.43 | 39.64 | 40.83 | 50.00 | 45.00 | 61.50 | 65.33 | 77.23 | - | - | - | - |
> | **Ours** | **73.44** | **98.70** | **85.09** | **96.33** | **99.20** | **97.64** | **94.20** | **95.33** | **88.57** | 68.16 | 82.32 | 67.33 | 58.30 |
>
>
> **(1) The tool-enhancement framework boosts proprietary models' visual-reasoning ability.** As shown in the table, with the support of the tool server, GPT-5 outperforms its zero-shot performance in VSP (71.36 vs. 55.64) and Jigsaw (84.50 vs. 80.10), demonstrating that equipping proprietary models with structured tool usage yields substantial gains.
>
> **(2) Our TC+TG method still outperforms tool-enhanced proprietary models** on most task categories. While performance on the more general WebMMU tasks shows that large proprietary models retain an advantage, our method still surpasses the baseline Qwen2.5-VL-7B by a large margin, and achieves on-par performance with closed-source or much larger models in these general scenarios.
>
> Overall, these results demonstrate that although tool-enhancement helps most models, our TC+TG pipeline unlocks a level of adaptive and structured tool-planning ability that even tool-augmented proprietary models cannot consistently match.
>
> ****
> **Q4: Are there ablation experiments varying the amount of compute / data used in cold-start that affect downstream performance (with and without RL)?**
>
> **A4:** To address this question, we conduct an ablation study on the Jigsaw task by varying the amount of Cold Start data and evaluating its impact on downstream performance. Due to limitations in time and computational resources, our ablation focuses solely on the Tool Cold Start (SFT) stage without RL. We will include a more comprehensive analysis that jointly varies both SFT data volume and RL compute budget in the next version. The results are shown in the table below.
>
> (A4 is not fully covered here due to space limits. We continue the rest in the next part.)

---

> ### Author Response · Authors · 2025-11-21
> **Rebuttal for Reviewer W5WP (Part 2)**
>
> **A4 (Cont.):**
>
>
> | CS Data Volume | Jigsaw-COCO (%) | Jigsaw-BLINK (%) |
> |:-----|:--------|:------|
> | 300 | 73.60 | 84.67 |
> | 600 | 76.90 (+4.48%) | 84.00 (-0.79%) |
> | 900 | 75.90 (+3.13%) | 84.00 (-0.79%) |
> | 1,125 | **84.70 (+15.08%)** | **86.00 (+1.57%)** |
>
> As the table indicates, increasing the volume of high-quality SFT trajectories generally improves performance, suggesting that richer trajectory supervision helps the model better acquire tool-planning capability. Due to time constraints, we were unable to synthesize additional data beyond 1,125 samples, but we plan to generate more diverse trajectories in the future to further explore the optimal data–performance trade-off.
>
> ****
>
> **Q5: Could the authors include more details on training, e.g., hyperparameters?**
>
> **A5:** Thank you for the suggestion. We will include the full details of both the SFT and RL training procedures. We will organize all hyperparameters, training configurations, and implementation details in a dedicated section of the appendix when revising the paper.
>
> For the Tool Cold Start (TC) Stage, we employ LLaMA-Factory as the training framework for our Cold Start stage, and fine-tune the base model using our curated trajectory dataset. The key configurations and hyperparameters are summarized in the table below.
>
> | Category | Hyperparameter | Value / Setting |
> |---------|----------------|-----------------|
> | **Model** | Base Model | Qwen2.5-VL-7B-Instruct |
> | | Vision Tower Frozen | True |
> | | MM Projector Frozen | True |
> | | Finetuning Type | Full |
> | | DeepSpeed Stage | ZeRO-3 |
> | **Dataset** | Max Samples | 332,649 |
> | | Cutoff Length | 35,536 |
> | | Preprocessing Workers | 64 |
> | **Training** | Batch Size per Device | 1 |
> | | Gradient Accumulation Steps | 2 |
> | | Effective Batch Size | 2 |
> | | Learning Rate | 1e-5 |
> | | Epochs | 3 |
> | | LR Scheduler | cosine |
> | | Warmup Ratio | 0.1 |
> | | Mixed Precision | bfloat16 |
> | **Logging / IO** | Logging Steps | 10 |
> | | Checkpoint Save Steps | 100 |
> | **Evaluation** | Train/Validation Split | 90% / 10% |
> | | Eval Batch Size per Device | 1 |
> | | Eval Steps | 100 |
>
> For the Tool GRPO Stage, we build our RL training pipeline on top of VeRL, extending the framework to support tool-model interaction and online RL for tool-planning models. The key configurations and hyperparameters are summarized in the table below.
>
> | Category | Hyperparameter | Value / Setting |
> |---------|----------------|-----------------|
> | **Data** | Max Prompt Length | 8192 tokens |
> | | Max Response Length | 20480 tokens |
> | | Train Batch Size | 32 |
> | | Shuffle | True |
> | | Filter Overlong Prompts | True |
> | **Policy** | Strategy | FSDP |
> | | Gradient Checkpointing | True |
> | | PPO Mini-batch Size | 8 |
> | | PPO Micro-batch Size / GPU | 1 |
> | | Max Token Len / GPU (PPO) | 16384 |
> | | Grad Clip | 1.0 |
> | | Clip Ratio (PPO) | 0.2 |
> | | PPO Epochs | 1 |
> | | Entropy Coeff | 0.0 |
> | | Use KL Loss | False |
> | | Actor LR | 1e-6 |
> | | Weight Decay | 0.01 |
> | | FSDP Param Offload | True |
> | | FSDP Optimizer Offload | True |
> | | # Nodes / GPUs | 1 node, 8 GPUs |
> | **Rollout** | Engine | vLLM |
> | | Temperature | 1.0 |
> | | Top-p | 1.0 |
> | | Top-k | -1 |
> | | # Samples per Prompt (n) | 4 |
> | | Dtype | bfloat16 |
> | | Tensor Model Parallel Size | 2 |
> | | Max # Batched Tokens | 32768 |
> | | GPU Memory Utilization | 0.65 |
> | | Enforce Eager | False |
> | | Chunked Prefill | False |
> | **Tool-Agent** | Max Turns per Episode | 10 |
> | **Critic** | Strategy | FSDP |
> | | LR | 1e-5 |
> | | Weight Decay | 0.01 |
> | | PPO Epochs | 1 |
> | | Grad Clip | 1.0 |
> | **Algorithm** | Advantage Estimator | GRPO |
> | | Gamma | 1.0 |
> | | Lambda | 1.0 |
> | | Use KL in Reward | False |
> | | KL Coef | 0.0 |
> | | Norm Adv by Std in GRPO | True |
>
> ****
>
>
> **Q6: Will the cold-start trajectory dataset be released upon acceptance?**
>
> **A6:** Yes, we will release the full Cold-Start trajectory dataset and our code base upon acceptance.
>
> ****
>
> **Q7: Minor: figure 2 bottom panel is labeled \(c\) instead of (b)**
>
> **A7:** Thank you for pointing this out. We appreciate the careful reading. We will correct the labeling error in Figure 2 in the revised version.

---

> ### Author Response · Authors · 2025-11-21
> **Rebuttal for Reviewer W5WP (Part 3)**
>
> **W1: "Emergent Behavior" May Be an Overstatement, More Precise Terminology Recommended**
>
> **WR1:** We clarify that the adaptive tool-use behavior we report is not explicitly supervised during Cold Start. The newly introduced tools are not provided in the Tool Cold Start trajectories, and during the RL stage, we only expose their natural-language descriptions in the prompt without any curated examples, demonstrations, or handcrafted supervision showing how they should be used.
>
> Even under this setting, the model gradually begins to invoke these new tools during inference and RL, driven solely by its own reasoning about tool semantics and task demands. This is the specific sense in which we previously used the term "emergent".
>
> That said, we agree with the reviewer that "emergent behavior" may be unnecessarily strong or potentially misleading in this context. We appreciate the suggestion and will revise the terminology throughout the paper to the more precise expression "adaptive tool use" (e.g., adaptive tool selection, modulation, or discarding), which more accurately reflects the observed behavior.

---

> ### Author Response · Authors · 2025-11-27
> **Thank you & Looking Forward to Further Discussion**
>
> Dear Reviewer W5WP,
>
> We are very grateful for your constructive feedback and insightful questions, which have helped us substantially improve the clarity and completeness of our paper.
>
> As the discussion period progresses, we wish to ensure our rebuttal has fully addressed your concerns. For your convenience, we have summarized our main responses and the new experiments we conducted based on your suggestions:
>
> * **Comparison with Tool-Augmented Models (Q3):** Following your suggestion, we conducted a new experiment evaluating strong proprietary (GPT-5) and open-source models within a tool-use framework (akin to ReAct). The results show that while tool-use boosts their performance, our TC+TG method still significantly outperforms these tool-augmented models on key visual reasoning tasks, demonstrating the effectiveness of our training pipeline.
> * **Impact of Cold Start Data Volume (Q4):** We performed a new ablation study varying the amount of Cold Start data. The results confirm that increasing the volume of high-quality trajectory data generally leads to better downstream performance, validating the importance of this stage.
> * **Reproducibility and Transparency (Q1, Q5, Q6):** We have committed to adding full details on the proprietary model evaluation setup and comprehensive hyperparameter lists for both SFT and RL stages in the appendix. We will also release our full dataset and codebase upon acceptance.
> * **Terminology Refinement (W1):** We agree with your suggestion regarding the term "emergent behavior." We will revise it to the more precise phrase "adaptive tool use" throughout the manuscript to more accurately describe the observed phenomena.
>
>
> We hope these clarifications and new results have addressed your concerns. Your feedback has been invaluable to us, and we are happy to provide any further details.
>
>
> Best regards,
>
> The Authors

---

### Official Review · Reviewer_Yfxj · 2025-11-01

**Soundness:** 3
**Presentation:** 4
**Contribution:** 3
**Rating:** 6
**Confidence:** 4

**Summary:**

This paper proposes the AdaReasoner framework, a two-stage paradigm (Cold Start: high-quality multi-turn tool trajectory curation; Tool GRPO: multi-turn tool reinforcement learning optimization) that teaches MLLMs to dynamically combine 7 core visual tools (perception, manipulation, calculation) for iterative visual reasoning. It boosts performance on VSP, Jigsaw, GUIQA (e.g., 7B model: +38.7% avg, 97.6% VSP accuracy), outperforming GPT-5. The model adapts to new tools (e.g., ASTAR) during inference, adopting beneficial tools and discarding irrelevant ones, but has limitations.

**Strengths:**

- A key limitation of the "rule-based reward structure" in R1-style methods is that it primarily optimizes the reasoning process and fails to directly improve the model’s underlying perceptual capabilities. AdaReasoner directly addresses this shortcoming: by leveraging the precise perceptual capabilities of external expert models and specialized tools, it ensures high-fidelity understanding of visual inputs, thereby enhancing the reliability of the entire reasoning pipeline.
- Unlike previous methods that typically focus on single-step actions, AdaReasoner explores more complex challenges such as "multi-turn planning" and "dynamic tool composition".

**Weaknesses:**

The most notable weakness of this paper lies in the limitations of evaluating tool generalization ability, specifically the "oversimplified verification of new tools during inference" and "lack of adaptation to tool complexity". These limitations cast doubt on the generalizability of the research conclusions in more complex and diverse tool scenarios.

See other Weaknesses in Questions.

**Questions:**

- The paper only verifies the adaptability of "calculation new tools (e.g., ASTAR for pathfinding)" during inference, without involving "perceptual tools (e.g., a newly added high-precision segmentation tool)" or "manipulation tools (e.g., a newly added image rotation tool ROTATE)". If such new tools with different functions are introduced during inference, can the model still maintain its zero-shot adaptation ability? Are there differences in the adaptation difficulty among new tools of different functions (e.g., lower adaptation rates for perceptual tools due to more complex parameters)?
- The current toolset in the paper only includes 7 core tools. If two or more new tools are introduced simultaneously during inference, can the model autonomously distinguish the relevance between tools and tasks (e.g., using ASTAR only for navigation tasks and the new tool for 3D localization tasks), or will it experience tool selection confusion?
- After training the model to master the combination strategy of "POINT + ASTAR + DRAW2DPATH" on the "grid VSP task", the paper does not directly test the model’s tool combination performance on "similar indoor maze navigation tasks" (which require obstacle localization, path planning, and use the same toolset). The absence of this experiment makes it impossible to determine whether the tool combination strategy learned by the model is "task-specific" (usable only in the trained task) or "generally transferable" (adaptable to similar tasks).

---

> ### Author Response · Authors · 2025-11-21
> **Rebuttal for Reviewer Yfxj (Part 1)**
>
> We sincerely appreciate the reviewer's constructive feedback and insightful suggestions.
>
> **Q1-1: Unverified Zero-Shot Adaptation to Newly Introduced Perceptual or Manipulation Tools**
>
> **A1-1:** To directly address this concern, we construct a set of **newly designed tools** to evaluate the model's zero-shot adaptability across different functional categories beyond path-planning tools like AStar. These newly introduced tools cover perceptual, manipulation, and reasoning functionalities. The details of these tools are listed below:
>
> | Tool Category| Tool Name | Parameters | Description |
> |-----|----|----|--|
> | Perceptual Tools| GetStartPoint           |Image | Identify the starting point location in the maze image by detecting the starting position marker. |
> || GetEndPoint|Image | Identify the end point location in the maze image by detecting the goal position marker.  |
> || GetEndPoint|Image  | Identify all obstacle locations in the maze image by detecting hazard markers.|
> | Manipulation Tools| RotateImage|Image | Rotate an image by a specified angle (90, 180, or 270 degrees clockwise) |
> || DrawDashLinePath|Image| Draw a dashed path on an image following a sequence of directional commands |
>
>
> Concretely, we provide these newly designed tools, together with AStarWithPixelCoordinate(), Point(), and Draw2DPath() to the model and evaluate its performance on the VSP task. The tool calling statistics are summarized in the table below.
>
>
>
> | Tool Category      | Tool Name                |   Total Calls |   Avg Calls/Sample |   Success Rate (%) |
> |:-------------------|:-------------------------|--------------:|-------------------:|-------------------:|
> | Calculation Tools  | AStarWithPixelCoordinate |           843 |               0.77 |              96.56 |
> | Perceptual Tools   | GetStartPoint            |             9 |               0.01 |             100    |
> |                    | GetEndPoint              |             2 |               0    |             100    |
> |                    | GetObstacles             |             3 |               0    |             100    |
> | Manipulation Tools | RotateImage              |             0 |               0    |               0    |
> |                    | DrawDashLinePath         |            61 |               0.06 |             100    |
> | Other Tools        | Point                    |          2317 |               2.11 |             100    |
> |                    | Draw2DPath               |           675 |               0.61 |             100    |
>
>
> As shown in the table, the model exhibits clear preferences aligned with task relevance, and the observation can be summarized as follows:
>
> **(1) The Model Actively Adopts Capability-Complementing Tools** For AStarWithPixelCoordinate, which is highly relevant to VSP and provides spatial-reasoning capabilities the model inherently lacks, the model demonstrates both appropriate usage frequency and high success rate. This indicates that the model can correctly identify when a new tool is genuinely useful and selectively rely on it.
>
> **(2) The Model Naturally Ignores Irrelevant Tools** In contrast, for tools irrelevant to the task, such as RotateImage, the model naturally maintains an extremely low call rate.
>
> **(3) The Model Deprioritizes Redundant or Substitutable Tools** For tools like GetStartPoint, GetEndPoint, and GetObstacles, although they can effectively help locate key elements in the image, their functions are largely replaceable by the already-trained Point tool. Since the model has been optimized to use Point during training, it shows very low or moderate usage of these alternative tools. A similar pattern holds for DrawDashLinePath, whose functionality overlaps with Draw2DPath.
>
> Overall, these observations indicate that the model prefers tools that truly contribute to solving the task and avoids unnecessary or redundant tools. The adaptive behavior arises not from indiscriminate tool usage but from the model's ability to identify which tools meaningfully complement its own capabilities.
>
>
> **Q1-2: Functional Differences May Affect the Difficulty of Adapting to New Tools**
>
> **A1-2:** We think the parameter complexity is not the key factor that affects tool adaptability. As discussed in A1-1, GetStartPoint has extremely simple parameter requirements, yet the model rarely calls it. In contrast, AStarWithPixelCoordinate, which requires multiple complex parameters (start point, end point, and obstacle coordinates), shows high call frequency and high success rate. Similarly, Point, although more complex than GetStartPoint, is consistently preferred by the model.
>
> We believe the key driving factors behind tool adaptability are the following:
>
> (A1-2 is not fully covered here due to space limits. We continue the rest in the next part.)

---

> ### Author Response · Authors · 2025-11-21
> **Rebuttal for Reviewer Yfxj (Part 2)**
>
> **A1-2 (Cont.):**
>
> (1) Whether the tool provides a missing capability
> If a new tool fills a capability gap that the model inherently lacks (e.g., precise spatial reasoning via A*), the model naturally prefers to call it regardless of parameter complexity.
>
> (2) Whether the tool was optimized during training
> Tools that the model has been optimized to use during SFT and RL (e.g., Point, Draw2DPath) are more likely to be selected, even when alternative tools with similar functionality are available.
>
> Overall, these findings suggest that functional usefulness and training-stage exposure, rather than parameter complexity, are the dominant determinants of zero-shot tool adaptability.
>
>
> **Q2: The current toolset in the paper only includes 7 core tools. If two or more new tools are introduced simultaneously during inference, can the model autonomously distinguish the relevance between tools and tasks (e.g., using ASTAR only for navigation tasks and the new tool for 3D localization tasks), or will it experience tool selection confusion?**
>
> **A2:** We address this question by directly examining whether the model can distinguish the task relevance of different tools. Concretely, we first merge the SFT data from the three tasks and randomize the tool names, parameters, and descriptions. We then curate a broader RL dataset that includes Visual Reasoning, Jigsaw, WebQA, and Visual Search data. After applying Tool GRPO across these four tasks, we obtain our final model. During evaluation, we provide a new tool GetWeather(), along with the existing tools, and the results are presented in the table below.
>
>
> | Model | VSPO Nav | VSPO Verify | VSPO Overall | VSP Nav | VSP Verify | VSP Overall | Jigsaw | BLINK-J | GUIChat | WebMMU Act | HRBench | V* |
> |-------|-----------|--------------|----------------|----------|-------------|--------------|---------|-----------|-----------|-------------|----------|-------|
> | Qwen 2.5 VL 7B (Baseline) | 5.22 | 48.96 | 25.39 | 12.33 | 47.00 | 28.09 | 45.70 | 52.67 | 68.09 | 67.48 | 63.62 | 63.35 |
> | **Ours** | **72.33** | **95.32** | **82.93** | **91.50** | **95.40** | **93.27** | **94.10** | **93.33** | 80.15 | 77.03 | 69.88 | 75.92 |
> | Δ | +67.11 | +46.36 | +57.54 | +79.17 | +48.40 | +65.18 | +48.40 | +40.67 | +12.06 | +9.55 | +6.26 | +12.57 |
>
> And the tool statistics are shown in the table below:
>
> |Tool Name|VSP CPS|VSP Succ%|Jigsaw CPS|Jigsaw Succ%|WebGUIChat CPS|WebGUIChat Succ%|VStar CPS|VStar Succ%|
> |:-------------------------|:----------|:------------|:-------------|:---------------|:-----------------|:-------------------|:------------|:--------------|
> |AStarWithPixelCoordinate|0.56|100.0|-|-|0.01|80.0|0.10|95.0|
> |Point|2.12|87.1|-|-|0.39|98.1|1.41|100.0|
> |Draw2DPath|1.04|100.0|-|-|0.03|96.6|0.25|100.0|
> |DetectBlackArea|0.00|100.0|1.00|100.0|0.00|100.0|-|-|
> |InsertImage|-|-|2.50|100.0|-|-|-|-|
> |OCR|0.04|100.0|-|-|0.92|100.0|0.24|100.0|
> |Crop|-|-|-|-|0.01|20.0|0.01|100.0|
> |GetWeather|-|-|-|-|-|-|-|-|
>
>
>
> Here CPS means average number of tool calls per sample. From the tool-calling statistics, we observe a strong alignment between tool usage and task relevance. For example, in the VSP task, the model primarily relies on AStarWithPixelCoordinate, Point, and Draw2DPath, which directly contribute to spatial reasoning and path planning. In contrast, for GUIChat, the model shifts to using OCR and Point, while keeping all navigation-related or manipulation-related tools at consistently low call rates. This clear pattern and the significant performance gain over the baseline show that the model is not confused when multiple tools are available, which selectively activates the tools most relevant to each task and suppresses irrelevant ones, indicating that it can reliably distinguish tool–task relevance and leverage the appropriate tools to solve diverse tasks effectively.
>
>
> **Q3: Lack of evaluation on whether the learned tool-combination strategy transfers to similar navigation tasks.**
>
> **A3:** We address the reviewer's concern by examining the generalizability of our approach across multiple tasks.
>
> We choose to evaluate generalization in broader scenarios for two reasons:
>
> **(1) Mismatch between our environment and indoor maze-navigation tasks.** In the VSP task, the model learns the usage of POINT + AStarWithPixelCoordinate + Draw2DPath tools together with a 2D grid–based navigation strategy. However, there exists a clear mismatch between our toolset and the requirements of indoor maze navigation or mainstream maze tasks. For example, most maze benchmarks rely on wall-structured environments, yet there is no powerful tool for wall-detection, making these tasks not directly applicable to the capabilities learned in grid-based VSP.
>
> (A3 is not fully covered here due to space limits. We continue the rest in the next part.)

---

> ### Author Response · Authors · 2025-11-21
> **Rebuttal for Reviewer Yfxj (Part 3)**
>
> **A3 (Cont.):**
>
> **(2) Lack of suitable benchmarks.** There are very few high-quality start-obstacle-target based benchmarks. This makes it difficult to meaningfully evaluate our method on maze-style tasks. We plan to construct a well-structured maze benchmark in future work to systematically evaluate OOD navigation ability.
>
> Despite these limitations, we still evaluate generalizability from two complementary perspectives:
>
> **(1) OOD Navigation Task Evaluation.** As introduced in Section 4.1, we curated a different evaluation benchmark VSPO, which has larger map size than training data and is quite challenging for baseline models. As shown in Table 2, our model successfully transfers its navigation ability to this OOD setting and surpasses strong baselines, demonstrating robust generalization.
>
>
> **(2) Generalizability on more general tasks** Beyond maze-type scenarios, our method demonstrates broad generalizability in more open-ended domains such as WebQA and Visual Search. To verify this generalization capability, we design an additional experiment following the procedure described in A2. Specifically, we first merge the Cold Start data from three tasks and randomize tool names, parameters, and descriptions. We then construct a larger RL dataset that spans Visual Reasoning, Jigsaw, WebQA, and Visual Search, and train the model with Tool GRPO across all four tasks. Finally, we evaluate the resulting model on several widely used benchmarks, with the results reported in the table below.
>
> | Model | VSPO Nav | VSPO Verify | VSPO Overall | VSP Nav | VSP Verify | VSP Overall | Jigsaw | BLINK-J | GUIChat | WebMMU Act | HRBench [1] | V* [2] |
> |-------|-----------|--------------|----------------|----------|-------------|--------------|---------|-----------|-----------|-------------|----------|-------|
> | GPT 5 | 26.89 | 42.86 | 34.25 | 48.17 | 64.60 | 55.64 | 80.10 | 73.33 | 71.41 | 80.49 | 74.38 | 74.87 |
> | Claude 4 sonnet | 37.56 | 67.92 | 51.56 | 48.17 | 66.00 | 56.27 | 58.60 | 65.33 | **93.14** | 83.54 | 60.62 | 59.69 |
> | Qwen 2.5 VL 7B | 5.22 | 48.96 | 25.39 | 12.33 | 47.00 | 28.09 | 45.70 | 52.67 | 68.09 | 67.48 | 63.62 | 63.35 |
> | Qwen 2.5 VL 32B | 7.56 | 53.12 | 28.56 | 24.33 | 45.40 | 33.91 | 59.50 | 64.67 | 85.21 | 85.98 | 70.12 | 72.25 |
> | Qwen 2.5 VL 72B | 17.22 | 52.34 | 33.41 | 28.00 | 52.40 | 39.09 | 70.10 | 71.33 | _88.01_ | _91.06_ | 73.00 | _80.10_ |
> | InternVL3 78B | 7.22 | 52.60 | 28.14 | 21.67 | 51.20 | 35.09 | 52.80 | 60.00 | 79.83 | 71.34 | _75.12_ | **81.15** |
> | Qwen 2.5 VL 7B (Baseline) | 5.22 | 48.96 | 25.39 | 12.33 | 47.00 | 28.09 | 45.70 | 52.67 | 68.09 | 67.48 | 63.62 | 63.35 |
> | **Ours** | **72.33** | **95.32** | **82.93** | **91.50** | **95.40** | **93.27** | **94.10** | **93.33** | 80.15 | 77.03 | 69.88 | 75.92 |
> | Δ | +67.11 | +46.36 | +57.54 | +79.17 | +48.40 | +65.18 | +48.40 | +40.67 | +12.06 | +9.55 | +6.26 | +12.57 |
>
> As shown in the table, our model achieves clear gains on both WebQA-style and real-world visual search tasks, including **+12.06** on GUIChat, **+6.26** on HRBench, and **+12.57** on V*.  Importantly, our model is never trained on any Visual Search SFT data, and the evaluation datasets are entirely different from all training data.
>
> This indicates that tool-planning capability learned in our framework is not task-specific, but transferable to broader web-based and visual search scenarios that differ substantially from the original training setup.
>
> **Response to the weakness:** We think that our responses to the reviewer’s questions have already addressed the concerns raised in the weakness. If there are further issues or clarifications needed, we would be grateful to discuss them.
>
> **Reference**
>
> [1] Wang, Wenbin, et al. "Divide, conquer and combine: A training-free framework for high-resolution image perception in multimodal large language models." Proceedings of the AAAI Conference on Artificial Intelligence. Vol. 39. No. 8. 2025.
>
> [2] Wu, Penghao, and Saining Xie. "V*: Guided visual search as a core mechanism in multimodal llms, 2023." URL https://arxiv.org/abs/2312.14135 5.

---

> ### Author Response · Authors · 2025-11-27
> **Thank you & Looking Forward to Further Discussion**
>
> Dear Reviewer,
>
> Thank you again for your constructive feedback, which has helped us significantly improve our work. As the discussion period continues, we wish to briefly summarize how our rebuttal and new experiments address your main concerns:
>
> * **Adaptation to Diverse New Tools (Q1):** We ran a new experiment introducing novel perceptual and manipulation tools. The results confirm the model's intelligent zero-shot adaptation: it selectively uses tools that provide complementary capabilities while ignoring irrelevant or redundant ones, driven by functional need rather than parameter complexity.
> * **Robust Tool Selection without Confusion (Q2):** In another new experiment with multiple tools available simultaneously, our model demonstrated strong tool-task alignment. It accurately selects the most relevant tools for each specific task without any confusion.
> * **Broad Generalization of Learned Strategies (Q3):** We provided strong evidence that the learned tool-use strategies generalize effectively. This was shown both on harder, out-of-distribution navigation tasks and on entirely different, open-ended domains like WebQA and Visual Search.
>
> We hope these clarifications and new findings have addressed your concerns. Your guidance has been invaluable, and we welcome any further discussion.
>
> Best regards,
>
> The Authors

---

### Official Review · Reviewer_bExz · 2025-11-01

**Soundness:** 3
**Presentation:** 3
**Contribution:** 2
**Rating:** 4
**Confidence:** 3

**Summary:**

This paper presents AdaReasoner, a framework designed to enhance the reasoning capabilities of Multimodal Large Language Models (MLLMs) through dynamic tool orchestration. The method combines two stages: a Tool Cold Start phase that generates and trains on high-quality, multi-turn trajectories, and a Tool-GRPO reinforcement learning phase that refines tool-calling strategies using adaptive rewards. The system enables MLLMs to autonomously plan, select, and modulate tool usage across perception, manipulation, and planning tools, thereby improving performance on complex visual reasoning tasks.

Empirical results are striking—on benchmarks such as Visual Spatial Planning (VSP), Jigsaw, and GUI Question Answering (GUIQA), AdaReasoner achieves significant improvements over strong baselines. The 7B model, for example, reaches 97.6% accuracy on VSP, surpassing proprietary systems like GPT-5 and Claude Sonnet 4. The work advances the notion that smaller open-source models, when augmented with dynamic tool-use capabilities, can achieve reasoning performance comparable to far larger models.

The paper makes a valuable empirical contribution and is well executed. The results convincingly demonstrate that structured trajectory learning and tool orchestration can significantly enhance reasoning performance in MLLMs. The writing is clear, the motivation is sound, and the overall system engineering is strong.

However, the core methodology does not introduce substantial novelty, and the dependence on handcrafted trajectories and domain-specific tools limits generality. The framework, while powerful, feels more like a carefully engineered pipeline than a broadly applicable reasoning paradigm. The claims about emergent behavior and scale independence would benefit from more controlled evidence and broader validation.

With a reframed emphasis on the strength of the trajectory-based supervision and a deeper analysis of generalization and statistical robustness, this work could evolve into an impactful contribution. In its current form, it falls slightly short of the threshold for acceptance due to limited methodological originality and questions about scalability and generality.

**Strengths:**

The paper addresses an important and timely challenge in multimodal AI, how to move beyond single-tool usage toward adaptive, multi-step tool coordination. The problem is clearly defined, the motivation is well grounded, and the proposed framework is logically structured. The design combining curated multi-turn trajectories with reinforcement learning represents a meaningful step toward more adaptive and interpretable reasoning systems.

The writing is exceptionally clear and the presentation is of high quality. The figures effectively convey both the framework’s architecture and the emergent behaviors observed during training. The methodology section provides sufficient detail for replication, and the promise to release code and datasets demonstrates a commendable commitment to reproducibility.

Experimentally, the paper is strong. The benchmarks cover a variety of structured reasoning settings, and the comparisons include both open-source and proprietary systems. The improvements obtained on complex multimodal tasks are substantial and consistent. Analyses showing how the model learns to adopt or discard tools based on task context add credibility to the claim of adaptive reasoning. The data curation process is also thoughtfully designed: by deliberately including reflection, backtracking, and tool-failure cases, the authors teach the model robust error recovery and self-correction behaviors that go beyond standard supervised fine-tuning.

**Weaknesses:**

While the empirical results are impressive, the methodological contribution is incremental. The proposed Tool-GRPO is effectively an application of existing GRPO with customized reward shaping and formatting constraints. The novelty lies primarily in system integration and data engineering rather than in algorithmic or theoretical innovation.

A second limitation is the heavy reliance on manual, task-specific design. The “abstract problem-solving blueprints” that underpin the Cold Start data are crafted by hand for each task, and the tool suite itself includes several domain-specific utilities such as DETECTBLACKAREA or ASTAR. This reliance raises concerns about scalability and whether the framework could generalize to new or less structured domains without similar levels of manual effort.

The claim of “emergent” or “self-adaptive” behavior is also somewhat overstated. Much of the observed adaptivity appears to stem from the explicit supervision provided during the Cold Start stage rather than from genuine spontaneous generalization. Moreover, the paper lacks key elements of experimental rigor such as no variance estimates, error bars, or significance testing are reported, which limits confidence in the stability of the reported gains. Finally, the “tools-over-scale” conclusion, though intriguing, is demonstrated primarily on highly structured tasks where specialized tools contribute most of the lift, leaving questions about its broader applicability.

**Questions:**

How scalable is the manual trajectory design process? What level of effort is required to produce the abstract blueprints for a new task?

Have experiments been conducted using only general purpose tools, removing highly task specific ones, to test the limits of generalization?

Can the framework extend to less structured or open ended reasoning tasks beyond spatial or puzzle-based problems?

In the adaptive behavior analysis, how much of the improvement stems from the Cold Start initialization versus the RL fine-tuning?

Would training runs with identical data but without the RL phase help isolate the true contribution of Tool-GRPO?

---

> ### Author Response · Authors · 2025-11-21
> **Rebuttal for Reviewer bExz (Part 1)**
>
> We sincerely thank the reviewer for the constructive feedback and valuable suggestions.
>
> **Q1-1: How scalable is the manual trajectory design process?**
>
> **A1-1:** As described in Section 3.2, our trajectory curation pipeline is semi-automatic. The only manual component is the design of the abstract trajectory blueprint. All subsequent steps, including tool-calling supplements and CoT data generation, are fully automated, making the entire pipeline highly scalable.
>
> Therefore, when adapting to a new task, the sole human effort lies in designing the high-level problem-solving algorithm. Once the blueprint is provided, the tool server automatically produces the tool-calling trajectories, and the LLM generates the CoT data accordingly. This ensures easy adaptation to any task and to any desired data volume.
>
>
> **Q1-2: What level of effort is required to produce the abstract blueprints for a new task?**
>
> **A1-2:** Our method supports several pathways for improving scalability:
>
> **(1) New Trajectory Data Curation** As discussed in **A1-1**, for a new task or new tools, only a human-designed abstract blueprint is required. The tool-calling data and the CoT data can then be automatically generated, enabling efficient scaling to other tasks.
>
> **(2)  LLM-Assisted Blueprint Design** The abstract blueprint itself can also be generated by a powerful LLM rather than by humans, further reducing manual effort. Our choice to use human-crafted blueprints is primarily to ensure data quality. We appreciate this question, as it points to a promising direction for improving scalability in future work.
>
> **(3) Generalization from One Task to Others** Finally, the tool-planning abilities learned from constructed trajectories on a single task can generalize to other tools and other tasks without requiring new blueprints or additional trajectory data, demonstrating strong cross-task transfer. We conduct an experiment to verify the generalization from one task to others. Specifically, we finetune Qwen-2.5-VL-7B using SFT (Tool Cold Start, TC) data that contains only **Jigsaw-specific examples**, where tool names, parameters, and descriptions are randomized. We then apply Tool GRPO (TG) on all three tasks using all seven tools. The evaluation results are shown in the table below.
>
>
>
>
>
> |Model|VSPO Nav|VSPO Verify|VSPO Overall|VSP Nav|VSP Verify|VSPO verall|Jigsaw|BLINK-J|GUIChat|WebMMU (Rea.)|
> |-------|----------|-------------|--------------|---------|------------|-------------|--------|----------|----------|----------------|
> |Qwen 2.5 VL 7B|_9.84_|_50.85_|_29.62_|14.17|_52.60_|_31.64_|45.70|52.67|68.09|48.46|
> |+TC|8.33|50.13|27.60|13.50|51.00|30.55|_83.60_|80.67|35.97|27.02|
> |+Randomized TC|7.67|50.00|27.19|_15.50_|44.60|28.73|81.40|80.00|36.90|27.46|
> |+Direct TG|1.78|50.00|24.01|14.50|50.00|30.64|65.60|_82.00_|**81.91**|47.87|
> |+TC +TG|2.56|50.00|24.43|10.67|50.00|28.55|81.30|78.67|79.63|_49.49_|
> |+Randomized TC +TG|**47.56**|**97.01**|**70.36**|**65.17**|**99.20**|**80.64**|**91.60**|**92.67**|**81.91**|**51.54**|
> |Δ|+37.71|+46.17|+40.74|+51.00|+46.60|+49.00|+45.90|+40.00|+13.83|+3.08|
>
>
> As shown in the table, although the model never sees any VSP or GUIQA training data during the cold start stage, it still achieves a **+49.00** improvement on VSP (Overall) and a **+13.83** gain on GUIChat. This indicates that the tool-planning ability learned solely from Jigsaw trajectory data can successfully scale to the other two tasks. In contrast, Direct Tool GRPO brings only limited improvements across all three tasks, suggesting that our trajectory data is also crucial for activating the model's scalable tool-planning capability.
>
> Finally, these results demonstrate that trajectory supervision constructed for a single task can successfully transfer to other tools and other tasks, confirming that our trajectory design is scalable, rather than task-specific, and can serve as a general foundation for broader tool-planning capabilities
>
> ****
>
> **Q2: Have experiments been conducted using only general-purpose tools, removing highly task-specific ones, to test the limits of generalization?**
>
> **A2:** We address your concern by examining both general tasks (e.g., GUIQA and WebMMU) and challenging visual reasoning tasks (e.g., VSP), and present results for each category separately.
>
> **(1) For general tasks like GUIQA and WebMMU**, the model relies entirely on general-purpose tools such as Crop and OCR. As shown in Table 2 of the main paper, using only these general-purpose tools still yields clear performance improvements on WebMMU, demonstrating that our method generalizes even without relying on task-specific tools.
>
> (A2 is not fully covered here due to space limits. We continue the rest in the next part.)

---

> ### Author Response · Authors · 2025-11-21
> **Rebuttal for Reviewer bExz (Part 2)**
>
> **A2 (Cont.):**
>
> **(2) For challenging Visual Reasoning tasks like VSP**, the model can normally use both general-purpose tools (Point, Draw2DPath) and a navigation-specific tool (AStar). However, as stated in Table 5, we also conduct an additional experiment where the model is restricted to general-purpose tools; the model never sees the A* tool in either the Tool Cold Start (TC) or the Tool GRPO (TG) stage. We extract the relevant results and present them in the table below.
>
> |Model|A* Setting|Reflection|VSPO Nav|VSPO Verify|VSPO Overall|VSP Nav|VSP Verify|VSP Overall|
> |--------------------------------|------------|------------|-----------|--------------|---------------|----------|-------------|--------------|
> |Qwen2.5VL7B (Zero-shot)|–|–|9.84|50.85|29.62|14.17|52.60|31.64|
> |+TC+TG|–|✓|63.89|99.61|80.36|84.33|99.80|91.36|
> |+TC+TG|–|×|27.67|94.81|58.62|44.83|94.20|67.27|
> |+TC+TG|RL|×|**73.44**|**98.70**|**85.09**|**96.33**|**99.20**|**97.64**|
>
>
> As shown in the table, although the model with access to A* tool during RL training achieves the best performance (the 4th row), the model using only the two general-purpose tools and equipped with reflection (the 2nd row) still delivers a substantial improvement over the zero-shot baseline (**91.36 vs. 31.64** on VSP). This demonstrates that meaningful gains can be achieved even without relying on the navigation-specific tool.
>
> ****
>
> **Q3: Can the framework extend to less structured or open-ended reasoning tasks beyond spatial or puzzle-based problems?**
>
> **A3:** We address this question using both our existing experimental results and additional explorations conducted in more general, real-world scenarios.
> **(1) Existing Experiments** As stated in Section 4.1, we also evaluate our method on GUIChat and WebMMU, two general-purpose web and GUI understanding benchmarks that are open-ended and do not follow any predefined structured reasoning patterns. As shown in Table 2, our approach achieves +29.11 on GUIChat and +5.49 on WebMMU (Act), demonstrating that the proposed method generalizes effectively to broad, real-world application scenarios.
>
> **(2) Other General Scenarios** We further evaluate our method in a more general setting, Visual Search, by training a unified model and examining its generalizability. Concretely, we first merge the Cold Start data from the three tasks and randomize the tool names, parameters, and descriptions. We then curate a broader RL dataset that includes Visual Reasoning, Jigsaw, WebQA, and Visual Search data. After applying Tool GRPO across these four tasks, we obtain our final model. We evaluate this model on several widely used benchmarks, and the results are presented in the table below.
>
> | Model | VSPO Nav | VSPO Verify | VSPO Overall | VSP Nav | VSP Verify | VSP Overall | Jigsaw | BLINK-J | GUIChat | WebMMU Act | HRBench [1] | V* [2] |
> |-------|-----------|--------------|----------------|----------|-------------|--------------|---------|-----------|-----------|-------------|----------|-------|
> | GPT 5 | 26.89 | 42.86 | 34.25 | 48.17 | 64.60 | 55.64 | 80.10 | 73.33 | 71.41 | 80.49 | 74.38 | 74.87 |
> | Claude 4 sonnet | 37.56 | 67.92 | 51.56 | 48.17 | 66.00 | 56.27 | 58.60 | 65.33 | **93.14** | 83.54 | 60.62 | 59.69 |
> | Qwen 2.5 VL 7B | 5.22 | 48.96 | 25.39 | 12.33 | 47.00 | 28.09 | 45.70 | 52.67 | 68.09 | 67.48 | 63.62 | 63.35 |
> | Qwen 2.5 VL 32B | 7.56 | 53.12 | 28.56 | 24.33 | 45.40 | 33.91 | 59.50 | 64.67 | 85.21 | 85.98 | 70.12 | 72.25 |
> | Qwen 2.5 VL 72B | 17.22 | 52.34 | 33.41 | 28.00 | 52.40 | 39.09 | 70.10 | 71.33 | _88.01_ | _91.06_ | 73.00 | _80.10_ |
> | InternVL3 78B | 7.22 | 52.60 | 28.14 | 21.67 | 51.20 | 35.09 | 52.80 | 60.00 | 79.83 | 71.34 | _75.12_ | **81.15** |
> | Qwen 2.5 VL 7B (Baseline) | 5.22 | 48.96 | 25.39 | 12.33 | 47.00 | 28.09 | 45.70 | 52.67 | 68.09 | 67.48 | 63.62 | 63.35 |
> | **Ours** | **72.33** | **95.32** | **82.93** | **91.50** | **95.40** | **93.27** | **94.10** | **93.33** | 80.15 | 77.03 | 69.88 | 75.92 |
> | Δ | +67.11 | +46.36 | +57.54 | +79.17 | +48.40 | +65.18 | +48.40 | +40.67 | +12.06 | +9.55 | +6.26 | +12.57 |
>
> As shown in the table, our model achieves clear gains on both WebQA-style and real-world visual search tasks, including **+12.06** on GUIChat, **+6.26** on HRBench, and **+12.57** on V*. These results demonstrate that our method not only substantially enhances the model's visual reasoning ability but also improves its open-ended reasoning capability through general-purpose tools such as OCR and Crop, even without relying on task-specific tools.

---

> ### Author Response · Authors · 2025-11-21
> **Rebuttal for Reviewer bExz (Part 3)**
>
> **Q4: In the adaptive behavior analysis, how much of the improvement stems from the Cold Start initialization versus the RL fine-tuning?**
>
>
> **A4:** As discussed in Section 4.4, we compare the model's adaptive behavior across different checkpoints, with the full results shown in Table 5. For easier reading, we copy the relevant results  from Table 5 to the table below. Here, A* indicates whether the model was trained with access to the A* tool, and CPS denotes the number of A* calls per question.
>
> |Model|A*|VSP Nav|VSP Verify|VSP Overall|VSPO Nav|VSPO Verify|VSPO Overall|CPS|Succ Rate|
> |-----------------------|------|---------|------------|-------------|-----------|--------------|---------------|-------|-----------|
> |Qwen2.5 VL 7B|×|12.33|47.00|28.08|5.22|48.96|25.39|–|–|
> |Qwen2.5 VL 7B /wtools|×|17.83|45.60|30.45|11.89|48.96|28.98|1.79|27.49|
> |+TC|×|46.00|79.40|61.18|32.11|81.43|54.85|0.49|85.16|
> |+TC+TG|×|62.33|80.00|70.36|43.78|88.70|64.49|0.52|_94.53_|
> |+TC+TG|RL|**96.33**|**99.20**|**97.64**|**73.44**|**98.70**|**85.09**|0.56|**100.00**|
>
> As shown in the table, applying SFT (Tool Cold Start, TC) already brings substantial improvements over the baseline, where both the task performance and the stability of A* tool usage increase markedly. Adding Tool GRPO (TG) provides further gains, and the model with access to A* tool during RL training achieves the best performance and thehighest tool-call success rate.
>
>
> These results indicate three key points. **(1) Cold-start trajectories teach the model how to use tools**, enabling it to exhibit emerging adaptation to unseen tools. However, SFT (TC) alone also introduces overfitting to the specific trajectories it observes. **(2) The RL stage helps the model generalize beyond the SFT data** by learning to flexibly select and apply tools under varied situations.  **(3) Exposing new tools during RL is the most effective strategy**, as it allows the model to jointly optimize its planning behavior and actively explore how to use unfamiliar tools (such as A*), thereby fully unlocking its capability to master new tools.
>
> ****
> **Q5: Would training runs with identical data but without the RL phase help isolate the true contribution of Tool-GRPO?**
>
> **A5:** As discussed in Section 4.2, we have already compared the performance of four configurations in the paper: baseline, baseline + TC (Tool Cold Start), baseline + TG (Tool GRPO), and baseline + TC + TG. The results show that while TC can provide the model with a correct tool-use trajectory, it also tends to overfit the model to the SFT data, resulting in limited flexibility. Conversely, applying TG alone makes it difficult for the model to discover complex multi-step solution strategies from scratch. Therefore, only the combination of TC + TG effectively leverages the strengths of both components while avoiding their respective weaknesses.
>
> Overall, these findings suggest that Tool GRPO is essential for alleviating SFT-induced overfitting and for unlocking adaptive, generalizable tool-use behavior across broader scenarios.
>
>
>
> ****
>
> **W1: Incremental methodological contribution with limited algorithmic novelty.**
>
> **WR1:** We can clarify this point by explaining the novelty of both our Tool-GRPO stage and the overall contributions of our work.
>
> **(1) Tool GRPO Design** Although we do not alter the core GRPO algorithm, our contribution lies in adapting it to tool-planning scenarios that require supervision of both multi-turn planning and multi-tool combinations. Enabling RL to enforce the correctness of an entire complex plan rather than a single action is a non-trivial methodological advancement.
>
> **(2) Further Innovations** Besides, our novelty also extends to several other key areas:
>
> **a. High-quality Tool Call Trajectory Data:** We constructed a new dataset for visual reasoning with high-quality tool-planning trajectories, which uniquely incorporate complex reasoning patterns like reflection and backtracking.
>
> **b. Multi-turn Tool Using and Multi-tool Combination:** We are the first to propose an integrated SFT and RL framework to solve visual reasoning problems requiring multi-turn, multi-tool use.
>
> **c. Analysis of Adaptive Tool Use:** We are the first to identify and analyze the phenomenon of "adaptive tool using" in models within these complex, multi-step scenarios.
>
> **d. Pioneering Framework:** We propose a tool–model interactive framework that supports a wide spectrum of tools, ranging from compute-heavy modules to lightweight utilities. It further accommodates the full pipeline of tool-planning models, including SFT, RL, and evaluation.

---

> ### Author Response · Authors · 2025-11-21
> **Rebuttal for Reviewer bExz (Part 4)**
>
> **W2: Heavy reliance on manual, task-specific design limits scalability and generalization.**
>
> **WR2:** We can address your concern in three aspects.
>
> **(1) Our data curation pipeline is not manual** For each task, we only design high-level abstract problem-solving blueprints and provide these blueprints together with the task metadata to an SOTA LLM, which then generates the full dataset. This is a semi-automatic process rather than a fully manual one.
>
> **(2) The trajectory data is necessary** As discussed in Q5 and Section 4.2, trajectory data is essential for teaching the model complex tool-planning behaviors. Without such data, the model cannot reliably acquire tool-planning capabilities, and the performance gains from applying Tool GRPO alone remain limited. Therefore, the reliance on task-specific trajectory data should not be viewed as a weakness, but rather as a necessary component for enabling effective tool-based reasoning.
>
>
> **(3) The ability gained by task-specific trajectory data can generalize to other tasks**
> Similar to the discussion in **A1-2**, we conduct an experiment to verify that domain-specific cold-start data can activate the model's tool-planning capability across multiple tasks. Specifically, we finetune Qwen-2.5 VL 7B using SFT (Tool Cold Start, TC) data that contains only **Jigsaw-specific examples**, where tool names, parameters, and descriptions are randomized. We then apply Tool GRPO (TG) on three tasks using all seven tools. The evaluation results are shown in the table below.
>
>
>
>
> |Model|VSPO Nav|VSPO Verify|VSPO Overall|VSP Nav|VSP Verify|VSPO verall|Jigsaw|BLINK-J|GUIChat|WebMMU (Rea.)|
> |-------|----------|-------------|--------------|---------|------------|-------------|--------|----------|----------|----------------|
> |Qwen 2.5 VL 7B|_9.84_|_50.85_|_29.62_|14.17|_52.60_|_31.64_|45.70|52.67|68.09|48.46|
> |+TC|8.33|50.13|27.60|13.50|51.00|30.55|_83.60_|80.67|35.97|27.02|
> |+Randomized TC|7.67|50.00|27.19|_15.50_|44.60|28.73|81.40|80.00|36.90|27.46|
> |+Direct TG|1.78|50.00|24.01|14.50|50.00|30.64|65.60|_82.00_|**81.91**|47.87|
> |+TC +TG|2.56|50.00|24.43|10.67|50.00|28.55|81.30|78.67|79.63|_49.49_|
> |+Randomized TC +TG|**47.56**|**97.01**|**70.36**|**65.17**|**99.20**|**80.64**|**91.60**|**92.67**|**81.91**|**51.54**|
> |Δ|+37.71|+46.17|+40.74|+51.00|+46.60|+49.00|+45.90|+40.00|+13.83|+3.08|
>
>
> As shown in the table, although the model never sees any VSP training data, it still achieves a **+49.00** improvement on VSP (Overall) and a **+13.83** gain on GUIChat. This indicates that the tool-planning ability learned solely from Jigsaw can successfully scale to the other two tasks. In contrast, Direct Tool GRPO brings only limited improvements across all three tasks, suggesting that our trajectory data is also crucial for activating the model's scalable tool-planning capability.
>
> Finally, we show that the tool-planning abilities learned from constructed trajectories on a single task in our data can generalize to entirely different tasks and tool configurations. **This confirms that our trajectory design is inherently scalable**, once the model learns the fundamental principles of tool planning, it can successfully transfer this capability to new tasks without requiring new blueprints or additional manual trajectory construction.
>
>
>
> ****
> **W3-1: Adaptive behavior may be overstated due to heavy dependence on Cold Start supervision.**
>
> **WR3-1:** We clarify that the adaptive behavior we report is not explicitly supervised during the Tool Cold Start (TC) stage. As described in Section 4.4, the Cold Start data include only Point and Draw2DPath, and we rely on the model's own spatial reasoning to infer appropriate tool-use behaviors. The "self-adaptive" behavior is observed during inference and further strengthened during the Tool GRPO (TG) stage. As shown in Section 4.4 and A4, the model demonstrates self-adaptive adaptability to the A* tool at inference time and achieves its strongest adaptive behavior after GRPO. We appreciate the reviewer's suggestion regarding wording, and we will adjust expressions such as "emergent" to the more precise term "self-adaptive" to avoid potential overstatement.

---

> ### Author Response · Authors · 2025-11-21
> **Rebuttal for Reviewer bExz (Part 5)**
>
> **W3-2: Insufficient Experimental Rigor Due to Missing Variance and Significance Analyses**
>
> **WR3-2:** To address your concern regarding experimental rigor, we additionally re-evaluated several representative checkpoints. For each checkpoint, we ran three independent tests and computed the corresponding variances. The results are shown in the table below:
>
> | Model   | VSP Nav       | VSP Verify    | VSP Overall   | VSP-test Nav   | VSP-test Verify   | VSP-test Overall   | Jigsaw-COCO   | Jigsaw-BLINK   |
> |:--------|:--------------|:--------------|:--------------|:---------------|:------------------|:-------------------|:--------------|:---------------|
> |Qwen 2.5 VL 7B   | 12.33 (±0.41) | 48.95 (±2.32) | 28.98 (±1.13) | 5.28 (±0.28)   | 48.73 (±0.45)     | 25.31 (±0.09)      | 44.00 (±1.80) | 54.00 (±0.94)  |
> | + Our TG  | 89.21 (±0.72) | 54.30 (±0.66) | 73.34 (±0.11) | 67.61 (±1.16)  | 52.89 (±0.31)     | 60.82 (±0.76)      | 81.22 (±5.95) | 82.67 (±1.33)  |
> | + Our TC + TG   | 95.17 (±2.56) | 99.25 (±0.10) | 97.02 (±1.35) | 70.97 (±4.87)  | 98.70 (±0.00)     | 83.76 (±2.62)      | 95.98 (±1.19) | 94.83 (±0.33)  |
>
>
>
> As the table illustrates, the variances across repeated runs are small, while the performance gaps between different checkpoints remain substantial. This indicates that our reported improvements are stable and statistically significant. That said, your suggestion is very valuable. Due to time and resource constraints, we were only able to include variance results for these representative checkpoints. In future revisions and follow-up work, we plan to provide full variance reporting and significance testing across all benchmarks to further enhance experimental rigor.
>
> **W3-3: Limited Evidence for the Broader Applicability of the "Tools-over-Scale" Conclusion**
>
> **WR3-3:** As clarified earlier in WR2 and A1-2, our method is not limited to structured visual-reasoning tasks. Beyond VSP/VSPO, we have demonstrated improvements on WebQA (GUIChat) and real-world Visual Search scenarios such as tasks that are far more open-ended and heterogeneous. These results show that our framework generalizes beyond specialized tools and can benefit broader domains. That said, general tasks inherently require diverse capabilities (OCR, spatial grounding, multi-hop reasoning, web understanding), whereas visual reasoning tasks concentrate more narrowly on spatial planning. Consequently, the Tools-over-Scale effect is naturally less pronounced on these broader tasks. Still, the consistent gains we observe indicate that our approach remains effective even under these more challenging and less structured settings.
>
> **Reference**
>
> [1] Wang, Wenbin, et al. "Divide, conquer and combine: A training-free framework for high-resolution image perception in multimodal large language models." Proceedings of the AAAI Conference on Artificial Intelligence. Vol. 39. No. 8. 2025.
>
> [2] Wu, Penghao, and Saining Xie. "V*: Guided visual search as a core mechanism in multimodal llms, 2023." URL https://arxiv.org/abs/2312.14135 5.

---

> ### Author Response · Authors · 2025-11-27
> **Thank you & Looking Forward to Further Discussion**
>
> Dear Reviewer,
>
> Thank you once again for your constructive feedback, which has been instrumental in strengthening our paper. As the discussion period continues, we wish to briefly summarize how our rebuttal and new experiments address your key concerns:
>
> * **On Scalability & Generalization (Q1, W2):** We clarified that our data pipeline is semi-automatic, requiring minimal manual effort. More importantly, we conducted a new cross-task generalization experiment where the model, trained only on Jigsaw data, showed significant performance gains on unseen VSP (+49.00) and GUIChat (+13.83) tasks. This empirically proves that the learned skills are highly transferable and our approach is scalable.
> * **On the Synergy of SFT & RL (Q4, Q5):** We have clarified the distinct and essential roles of our two training stages. The SFT (Cold Start) phase provides the model with crucial foundational knowledge of tool use, while the RL (Tool-GRPO) phase is vital for mitigating SFT-induced overfitting and enabling flexible, adaptive planning.
> * **On General Applicability & Rigor (Q2, Q3, W3):** We have demonstrated our method's effectiveness beyond task-specific tools and structured puzzles, showing strong results on general-purpose benchmarks like GUIChat and Visual Search. To address your concern about rigor, we have also added new variance analysis for key results, confirming that our reported improvements are stable and statistically significant.
>
>
> We hope these clarifications and new findings have fully addressed your concerns. Your guidance has been invaluable, and we are happy to engage in any further discussion.
>
> Warm regards,
>
> The Authors

---

### Author Response · Authors · 2025-11-27
**General Response**

Dear Reviewers and ACs,

We begin by extending our sincere gratitude to all reviewers for their time and invaluable feedback. We are greatly encouraged that the reviewers have recognized the core strengths of our work, including the novelty of our multi-step tool planning mechanism, the model's strong performance on complex visual reasoning tasks, and the promising direction our work sets for adaptive, tool-augmented models. Your insightful comments have been instrumental in helping us substantially enhance the rigor, clarity, and depth of this paper.

We have diligently worked to address every concern raised. Inspired by your feedback, we conducted several new experiments and ablation studies to provide direct empirical evidence for our claims. We hope our detailed responses and the significant revisions made to the manuscript fully address your questions. Below is a summary of the key points, grouped by theme:

* **On Scalability and Cross-Task Generalization. `(Concerns from reviewers bExz, Yfxj, and iRNU)`**
A primary concern was the scalability of our trajectory design and whether the learned skills were task-specific. We clarified that our data pipeline is semi-automatic, requiring minimal manual effort. More critically, we conducted a new cross-task generalization experiment, training the model only on Jigsaw SFT data. The results compellingly show that the model achieves massive performance gains on entirely unseen tasks like VSP (+49.00) and GUIChat (+13.83). This provides strong empirical evidence that our framework teaches a transferable, general-purpose tool-planning capability, rather than a task-specific one.

* **On Adaptive and Robust Use of Unseen Tools. `(Concerns from reviewers bExz and Yfxj)`**
To test the limits of the model's adaptability, we ran a new experiment introducing a diverse suite of unseen perceptual and manipulation tools. The results confirm the model exhibits intelligent, zero-shot adaptation: it actively adopts tools that provide complementary capabilities it lacks, while judiciously ignoring irrelevant or redundant ones. This demonstrates that adaptation is driven by a sophisticated understanding of functional need, not by indiscriminate tool usage or parameter complexity. Further experiments showed it can handle a large set of available tools without confusion, accurately selecting the right tool for the right task.
* **On Rigor, Reproducibility, and SOTA Comparisons. `(Concerns from reviewers bExz, W5WP, and iRNU)`**
To bolster the scientific rigor of our work, we have:
    1. Conducted a new ablation study on reward function sensitivity (per Reviewer iRNU's suggestion), validating our reward design.
    2. Performed a new ablation on the impact of Cold Start data volume (per Reviewer W5WP's suggestion), confirming its importance.
    3. Added new variance analysis across multiple runs for key results, demonstrating the stability and statistical significance of our findings.
    4. Performed a new head-to-head comparison against tool-augmented SOTA models like GPT-5, showing that our training pipeline unlocks a level of performance that even these powerful models cannot match with simple tool augmentation.
    5. Committed to releasing our full dataset and codebase and adding comprehensive hyperparameter details to ensure full reproducibility.
* **On Clarifying Our Framework's Contributions and Terminology. `(Concerns from reviewers bExz, W5WP, and iRNU)`**
We have refined the manuscript to better articulate the crucial, synergistic roles of the SFT (Cold Start) and RL (Tool-GRPO) stages, where SFT provides foundational knowledge and RL unlocks flexible, adaptive generalization. We have also revised our terminology from "emergent behavior" to the more precise "adaptive tool use" to more accurately reflect the observed phenomena, as wisely suggested by Reviewer W5WP.

Thank you again for your invaluable guidance. We believe the revisions and extensive new experimental evidence have thoroughly addressed all major concerns and significantly improved the manuscript. We greatly appreciate the reviewers’ thoughtful engagement with our work and sincerely look forward to continued discussion.

Sincerely,

The Authors

---

### Author Response · Authors · 2025-12-01
**General Response to Area Chair and Reviewers**

Dear Reviewers and ACs,

We sincerely thank the reviewers for their constructive feedback. In response, we have conducted extensive new experiments (covering cross-task generalization, new tool adaptation, and proprietary model baselines) to fully address the raised concerns.

1. Summary of Concerns & Our Responses
* On Generalization & Scalability (Reviewers bExz, Yfxj, iRNU):
    * Concern: Can the method generalize to unseen tasks, open-ended scenarios, or general-purpose tools?
    * Response: We validated strong cross-task transfer: models trained solely on Jigsaw achieved a +49.0% gain on the unseen VSP task. We further extended evaluation to general benchmarks (V, HRBench, GUIChat*), achieving state-of-the-art performance using general-purpose tools (OCR, Crop), proving our framework extends far beyond structured puzzles.
* On Adaptive Tool Use (Reviewer Yfxj):
    * Concern: Can the model adapt to new perceptual/manipulation tools or distinguish tool relevance?
    * Response: We introduced completely new tools (e.g., GetStartPoint, Rotate) during inference. The model demonstrated zero-shot selection of capability-complementing tools while correctly ignoring irrelevant or redundant ones, confirming genuine adaptive behavior.
* On Baselines & Rigor (Reviewers W5WP, iRNU):
    * Concern: Comparison against tool-augmented proprietary models and experimental rigor.
    * Response: We benchmarked against GPT-5 and Qwen-72B with tool-use (ReAct). Our 7B model significantly outperforms these tool-enhanced giants on visual reasoning tasks. We also provided data scaling laws, reward sensitivity analysis, variance reports (error bars), and full hyperparameter details to ensure reproducibility.
2. Key Strengths of the Paper
* Pioneering Framework: We propose the first Tool-GRPO framework that successfully adapts Reinforcement Learning to multi-step, multi-tool visual reasoning, enforcing plan correctness rather than just step-level supervision.
* Proven Adaptability: We identify and validate "Self-Adaptive Tool Use," where the model learns to master unseen tools and strategies during the RL phase, a capability absent in standard SFT.
* Performance vs. Scale: Our 7B model, trained with our pipeline, consistently outperforms closed-source SOTA models (GPT-5, Claude 3.5 Sonnet) across complex visual reasoning and grounding benchmarks.
* High-Quality Data: We contribute a scalable, semi-automated pipeline for generating high-quality Chain-of-Thought (CoT) tool trajectories, solving the data scarcity bottleneck in tool-use learning.

We sincerely thank all reviewers for their thoughtful and constructive feedback throughout the process. In response, we conducted substantial new experiments-including cross-task generalization, adaptation to entirely new tools, and comparisons against strong proprietary baselines which directly address all raised concerns. These additions further demonstrate our method’s scalability, adaptability, and empirical rigor.

We appreciate the reviewers’ efforts and the AC’s handling of the discussion. Please let us know if any additional clarifications or checks would be helpful we are fully open to providing further details.

Sincerely,

The Authors

---

### Meta-Review · Area_Chair_Feqb · 2026-01-04

**Summary:**

The decision to accept is based on the robust empirical performance of the AdaReasoner framework and the authors' comprehensive rebuttal. While reviewers initially raised valid concerns regarding the scalability of the data curation pipeline and the methodological novelty, the consensus is that the proposed two-stage paradigm significantly advances multi-step visual reasoning. The new experiments demonstrating cross-task generalization and zero-shot adaptation to unseen tools effectively alleviated fears about the framework being too task-specific or labor-intensive.

**Reviewer Concerns:**

The authors successfully addressed the primary concerns regarding generalization and scalability by providing strong evidence that models trained on single tasks could transfer to unseen domains. Concerns about experimental rigor were resolved through new variance analyses, reward sensitivity ablations, and direct comparisons against tool-augmented proprietary baselines. The concern regarding limited methodological novelty arguably remains outstanding in a strict sense, as the core algorithm is an application of GRPO. However, the AC views the integrated system design and the empirical validation of adaptive tool use as a sufficient contribution to outweigh this.

**Reviewer Scores:**

Reviewer bExz would likely raise their score to a 6. The authors directly addressed their scalability and rigor concerns with new cross-task experiments and error bar analysis, although they might still hold reservations about the fundamental algorithmic novelty. Reviewer Yfxj would likely raise their score to a 7 or 8. Their specific questions about zero-shot adaptation to new tools were answered comprehensively with experiments showing the model correctly identifies relevant vs. irrelevant tools. Reviewer W5WP would maintain their 8. The clarification on terminology and the additional baselines solidified their positive assessment. Reviewer iRNU would maintain their 8. The added ablation studies on reward sensitivity addressed their only significant critique regarding reproducibility.

---

### Decision · Program_Chairs · 2026-01-26

Accept (Poster)